# Protein sequences bound to mineral surfaces persist into deep time

Beatrice Demarchi[1]*, Shaun Hall[2], Teresa Roncal-Herrero[3], Colin L Freeman[2]*, Jos Woolley[1], Molly K Crisp[4], Julie Wilson[4,5], Anna Fotakis[6], Roman Fischer[7], Benedikt M Kessler[7], Rosa Rakownikow Jersie-Christensen[8], Jesper V Olsen[8], James Haile[9], Jessica Thomas[6,10], Curtis W Marean[11,12], John Parkington[13], Samantha Presslee[1], Julia Lee-Thorp[9], Peter Ditchfield[9], Jacqueline F Hamilton[14], Martyn W Ward[14], Chunting Michelle Wang[14], Marvin D Shaw[14], Terry Harrison[15], Manuel Domínguez-Rodrigo[16], Ross DE MacPhee[17], Amandus Kwekason[18], Michaela Ecker[9], Liora Kolska Horwitz[19], Michael Chazan[20,21], Roland Kröger[3], Jane Thomas-Oates[4,22], John H Harding[2]*, Enrico Cappellini[6], Kirsty Penkman[4], Matthew J Collins[1]*

[1]BioArCh, Department of Archaeology, University of York, York, United Kingdom; [2]Department of Material Science and Engineering, University of Sheffield, Sheffield, United Kingdom; [3]Department of Physics, University of York, York, United Kingdom; [4]Department of Chemistry, University of York, York, United Kingdom; [5]Department of Mathematics, University of York, York, United Kingdom; [6]Centre for GeoGenetics, Natural History Museum of Denmark, University of Copenhagen, Copenhagen, Denmark; [7]Advanced Proteomics Facility, Target Discovery Institute, Nuffield Department of Medicine, University of Oxford, Oxford, United Kingdom; [8]Novo Nordisk Foundation Center for Protein Research, Faculty of Health Sciences, University of Copenhagen, Copenhagen, Denmark; [9]Research Laboratory for Archaeology and the History of Art, University of Oxford, Oxford, United Kingdom; [10]Molecular Ecology and Fisheries Genetics Laboratory, School of Biological Sciences, Bangor University, Bangor, United Kingdom; [11]Institute of Human Origins, SHESC, Arizona State University, Tempe, United States; [12]Centre for Coastal Palaeoscience, Nelson Mandela Metropolitan University, Port Elizabeth, South Africa; [13]Department of Archaeology, University of Cape Town, Cape Town, South Africa; [14]Wolfson Atmospheric Chemistry Laboratories, Department of Chemistry, University of York, York, United Kingdom; [15]Center for the Study of Human Origins, Department of Anthropology, New York University, New York, United States; [16]Department of Prehistory, Complutense University of Madrid, Madrid, Spain; [17]Department of Mammalogy, American Museum of Natural History, New York, United States; [18]National Museum of Tanzania, Dar es Salaam, Tanzania; [19]National Natural History Collections, Faculty of Life Sciences, The Hebrew University, Jerusalem, Israel; [20]Department of Anthropology, University of Toronto, Toronto, Canada; [21]Evolutionary Studies Institute, University of the Witwatersrand, Braamfontein, South Africa; [22]Centre of Excellence in Mass Spectrometry, University of York, New York, United States

*For correspondence: beatrice@palaeo.eu (BD); c.l.freeman@sheffield.ac.uk (CLF); j.harding@sheffield.ac.uk (JHH); matthew.collins@york.ac.uk (MJC)

**Competing interests:** The authors declare that no competing interests exist.

**Abstract** Proteins persist longer in the fossil record than DNA, but the longevity, survival mechanisms and substrates remain contested. Here, we demonstrate the role of mineral binding in preserving the protein sequence in ostrich (Struthionidae) eggshell, including from the

palaeontological sites of Laetoli (3.8 Ma) and Olduvai Gorge (1.3 Ma) in Tanzania. By tracking protein diagenesis back in time we find consistent patterns of preservation, demonstrating authenticity of the surviving sequences. Molecular dynamics simulations of struthiocalcin-1 and -2, the dominant proteins within the eggshell, reveal that distinct domains bind to the mineral surface. It is the domain with the strongest calculated binding energy to the calcite surface that is selectively preserved. Thermal age calculations demonstrate that the Laetoli and Olduvai peptides are 50 times older than any previously authenticated sequence (equivalent to ~16 Ma at a constant 10°C).

## Introduction

### Unknown mechanisms of survival of proteins into deep time

Ancient protein and DNA sequences are revolutionising our understanding of the past, providing information on phylogeny, migration, evolution, domestication and extinction (*Hagelberg et al., 2015*; *Cappellini et al., 2014*). However, the absence of data from warm regions and deep time (*Wade, 2015*) highlights the fragility of these biomolecules and has so far hampered our ability to answer fundamental evolutionary questions, such as resolving the phylogenetic tree of the genus *Homo* in Africa. The survival of proteins and DNA in tropical environments and in fossils that go back a few million years (Ma) is deemed extremely unlikely and therefore the impact of the 'biomolecular revolution' in palaeontology and palaeoanthropology has so far been relatively limited.

Claims for exceptional preservation in the fossil record have been put forward in a number of studies (*Towe and Urbanek, 1972*; *Bertazzo et al., 2015*; *Schweitzer et al., 2013*; *Cleland et al., 2015*), but these have not been satisfactorily substantiated. Morphological (*Towe and Urbanek, 1972*; *Bertazzo et al., 2015*), immunological (*Schweitzer et al., 2013*) and spectroscopic (*Bertazzo et al., 2015*) evidence of preserved tissues in dinosaurs and other fossils seems to be inconsistent with the observed levels of hydrolysis, dehydration and racemization (*Penkman et al., 2013*) in intracrystalline proteins from the fossil mollusc shell (*Sykes et al., 1995*) and eggshell (*Brooks et al., 1990*). The mechanisms that might allow preservation over palaeontological and geological time scales are also poorly understood: crosslinking, organo-metallic complexing, including with iron, compression/confinement (*Logan et al., 1991*; *Schweitzer et al., 2014*), and mineral stabilization (*Collins et al., 2000*) have all been proposed as mechanisms that enhance the survival of ancient biomolecules.

### The role of temperature in accelerating diagenesis

A confounding factor when evaluating the authenticity and antiquity of biomolecular sequences is the geographic area of provenance of the fossils and therefore the combined effect of time and temperature on the extent of degradation. Here we have used kinetic estimates of degradation rates of DNA (*Allentoft et al., 2012*), collagen in bones (*Buckley et al., 2008*), and intracrystalline amino acids (*Crisp et al., 2013*) to normalize their numerical (chronological) ages to thermal age (*Wehmiller, 1977*) (*Figure 1*, *Figure 1—source data 1*, Appendix 1). Thermal age is a measure which enables simple comparison between ancient biomolecular targets by normalising them to an equivalent (thermal) age, allowing all samples to be treated as having experienced a constant temperature of 10°C. Thus samples from cooler sites, which experience slower rates of chemical reaction, will have thermal ages younger than their geochronological age, whilst samples from warmer sites will be thermally 'older'. Various factors can affect the effective diagenetic temperature experienced by a sample (and therefore impact on its thermal age), from burial depth to seasonal and interglacial / glacial cycles (*Wehmiller, 1977*; *Huang et al., 2000*; *Eischeid et al., 1995*). The greatest absolute ages for recovered DNA (*Orlando et al., 2013*) (0.7 Ma = 0.08 Ma@10°C) and for protein (*Rybczynski, 2013*) (3.5 Ma = 0.3 Ma@10°C) sequences are from high latitudes and their survival is consistent with predictions from the kinetic data. Younger samples from more temperate latitudes will have greater thermal ages, yet the oldest of these which has preserved protein (Weybourne Crag: 1.5 Ma = 0.2 Ma@10°C) has a thermal age similar to that of Middle Pleistocene DNA at Sima de los Huesos (0.4 Ma = 0.2 Ma@10°C) (*Meyer et al., 2014*).

**eLife digest** The pattern of chemical reactions that break down the molecules that make our bodies is still fairly mysterious. Archaeologists and geologists hope that dead organisms (or artefacts made from them) might not decay entirely, leaving behind clues to their lives. We know that some molecules are more resistant than others; for example, fats are tough and survive for a long time while DNA is degraded very rapidly. Proteins, which are made of chains of smaller molecules called amino acids, are usually sturdier than DNA. Since the amino acid sequence of a protein reflects the DNA sequence that encodes it, proteins in fossils can help researchers to reconstruct how extinct organisms are related in cases where DNA cannot be retrieved.

Time, temperature, burial environment and the chemistry of the fossil all influence how quickly a protein decays. However, it is not clear what mechanisms slow down decay so that full protein sequences can be preserved and identified after millions of years. As a result, it is difficult to know where to look for these ancient sequences.

In the womb of ostriches, several proteins are responsible for assembling the minerals that make up the ostrich eggshell. These proteins become trapped tightly within the mineral crystals themselves. In this situation, proteins can potentially be preserved over geological time. Demarchi et al. have now studied 3.8 million-year-old eggshells found close to the equator and, despite the extent to which the samples have degraded, discovered fully preserved protein sequences.

Using a computer simulation method called molecular dynamics, Demarchi et al. calculated that the protein sequences that are able to survive the longest are stabilized by strong binding to the surface of the mineral crystals. The authenticity of these sequences was tested thoroughly using a combination of several approaches that Demarchi et al. recommend using as a standard for ancient protein studies.

Overall, it appears that biominerals are an excellent source of ancient protein sequences because mineral binding ensures survival. A systematic survey of fossil biominerals from different environments is now needed to assess whether these biomolecules have the potential to act as barcodes for interpreting the evolution of organisms.

## Aim of the study: understanding protein survival in ostrich eggshell from hot environments

Here we explore the impact of strong protein binding in biominerals and its effect on sequence survival, by targeting ancient ostrich eggshell (*Struthio camelus*; Struthioniformes), which is abundant in archaeological and palaeontological sites throughout Africa (Materials and methods). Our aim was to elucidate a mechanism of preservation and to set out a rigorous methodology for the authentication of ancient protein sequences. We isolated and characterised the intracrystalline proteins (*Figure 2*, *Figure 2—figure supplement 1*, Appendix 2) and tracked their diagenesis back in time to 3.8 Ma ago. Using a systematic approach, we validated the sequences from each of the eggshell samples analysed using amino acid racemization (AAR), organic volatile compounds, ancient DNA and proteomic analyses. All our results are supported by in-depth analysis of patterns of diagenesis in both samples and blanks as well as the evaluation of potential contamination and carry-over.

## Results

### Fossil eggshell from Africa, 0–3.8 Ma: provenance and thermal age calculations

Twenty-four eggshell samples were sourced from well-dated sites in South Africa and Tanzania (*Figure 1*, *Table 1*): Elands Bay Cave (0.3–16 ka BP, *Table 3*), Pinnacle Point Caves PP 5/6 and PP 30 (50–80 ka BP and ~150 ka BP, respectively, *Table 4*), Wonderwerk Cave (1 Ma, *Table 5*), Olduvai Gorge (1.34 Ma, *Table 6*) and Laetoli (2.6–4.3 Ma, *Table 7*). The age and stratigraphy of the oldest fossils, from Laetoli, is well-constrained despite the eggshell fragments being surface finds: their morphology shows no evidence of long-distance transport and the fossil-bearing horizons are well identified within the stratigraphy. The absence of lava flows in stratigraphic proximity also excludes

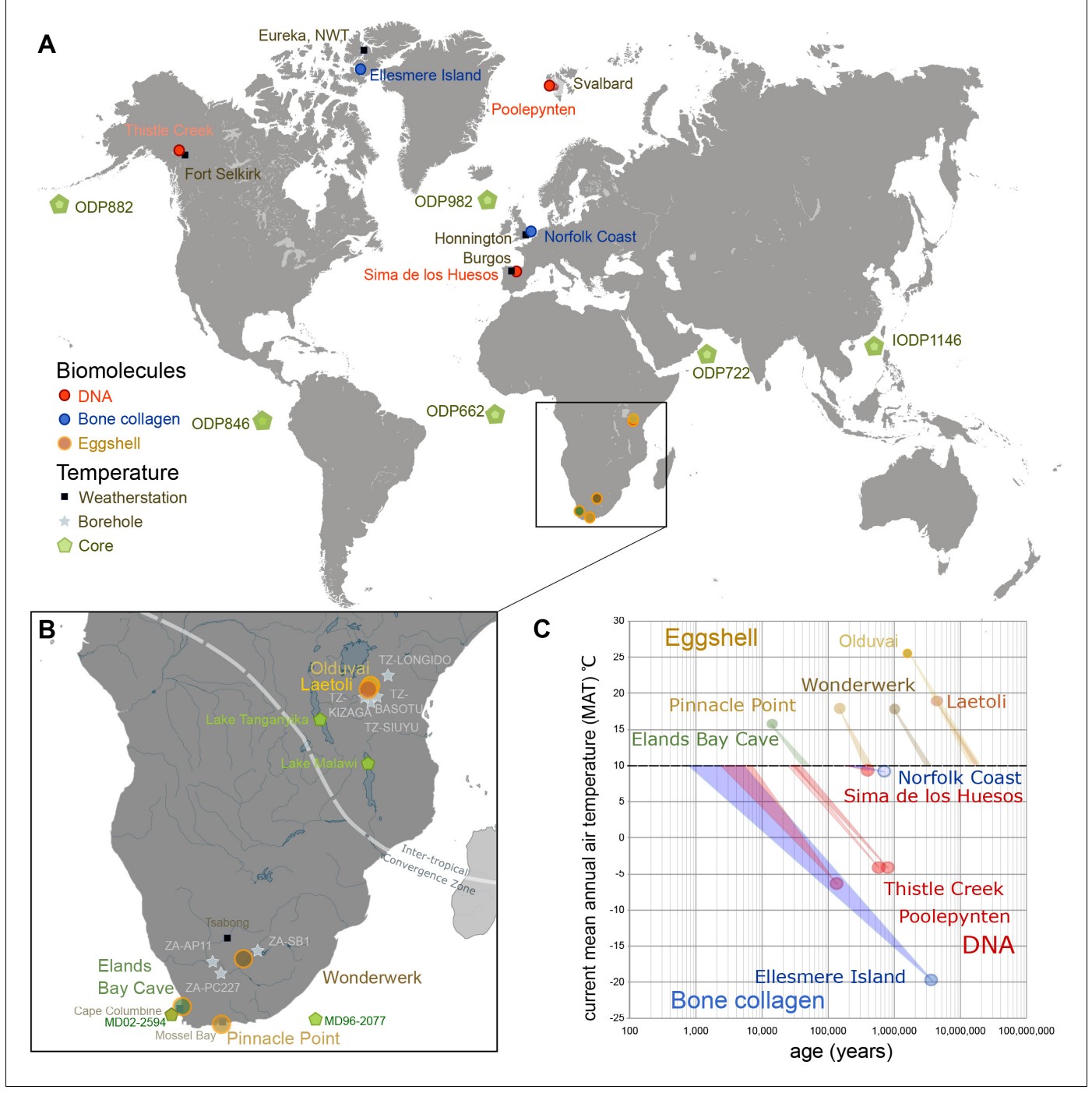

**Figure 1.** Eggshell peptide sequences from Africa have thermal ages two orders of magnitude older than those reported for DNA or bone collagen. (A) Sites reporting the oldest DNA and collagen sequences are from high latitude sites compared to ostrich eggshell samples from sites in Africa illustrated in (B) for which the current mean annual air temperatures are much higher. (C) Kinetic estimates of rates of decay for DNA (*Lindahl and Nyberg, 1972*), collagen (*Buckley and Collins, 2011*) and ostrich eggshell proteins (*Crisp et al., 2013*) were used to estimate thermal age assuming a constant 10°C (*Figure 1—source data 1*; Appendix 1. For Elands Bay Cave and Pinnacle Point the oldest samples are shown). Note log scale on the z-axis: struthiocalcin-1 peptide survival is two orders of magnitude greater than any previously reported sequence, offering scope for the survival of peptide sequences into deep time.

The following source data is available for figure 1:

**Source data 1.** Data and calculations for thermal ages reported in *Figure 1* and in *Supplementary file 1*.

the possibility that these fragments had been exposed to additional heat. The provenance of the fragments from Olduvai Gorge is also secure, as these were found *in situ* during the excavation of the Bell Korongo site, which overlies directly a volcanic tuff dated by $^{40}$Ar/$^{39}$Ar (*Domínguez-Rodrigo et al., 2013*).

The chronological ages of the samples were normalised to thermal ages: the mean annual air temperature (MAT) for each site was estimated from the NOAA NCDC GCPS monthly weather station (*Eischeid et al., 1995*; *Karl et al., 1990*) and borehole data (*Huang et al., 2000*; *National Climatic Data Center (NCDC), 2012*) (*Appendix 1—table 1*). Samples on the surface or buried at shallow depth will have experienced an effective temperature which is higher than the MAT, as rates of reaction scale exponentially with temperature (*Wehmiller, 1977*). The greater the seasonal range at the site, the older the thermal age will be, but the effect of seasonal fluctuations will be mitigated by burial depth, which dampens temperature changes. Holocene sites which today have a MAT of exactly 10°C will have been cooler in the past 500 years due to recent anthropogenic

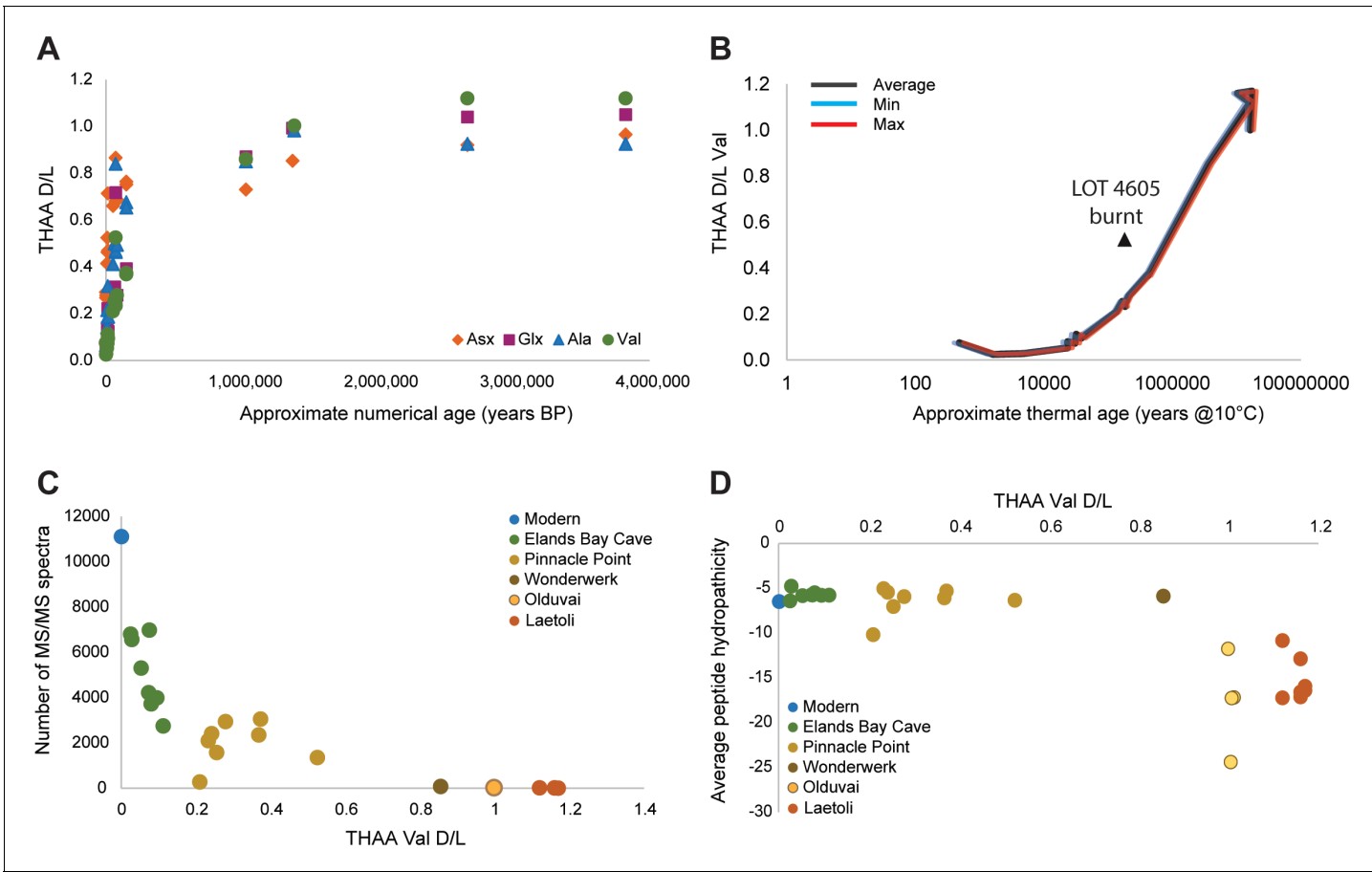

**Figure 2.** Proteome persistence and patterns of degradation. (**A**) Amino acid racemization in ostrich eggshell up to 3.8 Ma old from sites in South Africa and Tanzania. (**B**) Linear increase of THAA Val D/L values with the log of thermal age. (**C**) Exponential decrease of the number of identified MS/MS spectra with age (THAA Val D/L). (**D**) The average hydropathicity of the peptides identified remains stable up to Val D/Ls ~1. Note that Val D/Ls > 1.0 are unexpected and may be due to decomposition processes occurring in the closed system. The intracrystalline proteome composition in modern eggshell does not vary across microstructural layers (*Figure 2—figure supplement 1*), but patterns of degradation are different between fossil samples and purified proteins degraded at high temperature in the absence of the mineral (*Figure 2—figure supplement 2*).

The following figure supplements are available for figure 2:

**Figure supplement 1.** Structure and composition of OES.

**Figure supplement 2.** Proteome degradation.

**Table 1.** Summary of archaeological and paleontological eggshell samples analysed in this study. See also *Tables 3–7*.

| Site | Approximate age range (ka) | Approximate thermal age range (ka@10°C) | Number of specimens |
|---|---|---|---|
| Elands Bay Cave | 0.3–16 | 0.5–45 | 8 |
| Pinnacle Point 5/6 | 50–150 | 120–470 | 8 |
| Wonderwerk | ~1000 | ~3600 | 1 |
| Olduvai | ~1340 | ~16300 | 4 |
| Laetoli | 2600–4300 | 8900–20400 | 3 |

warming. In this study, we used borehole temperature estimates (*Huang et al., 2000*) or long-term historic records (*Eischeid et al., 1995*) to counter this effect. Pre-Holocene samples from sites which today have an MAT of 10°C will have an even younger thermal age due to the reduction in temperature during glacial periods. This retards the rate of chemical degradation, and therefore slows the advance of thermal age. While we did not correct for seasonal fluctuation, a correction was applied for altitude. The long-term temperature model of *Hansen et al. (2013)*, scaled to local or regional estimates of present day values and predicted temperature decline at the last glacial maximum (LGM), was used to project MATs from present day to the time of deposition (*Appendix 1—tables 1*, *2*, *4*).

Ostrich eggshell protein degradation was compared with the extent of degradation of DNA and bone collagen detected in a variety of Northern Hemisphere sites (*Figure 1*). Published kinetic parameters for the degradation of the molecules (*Appendix 1—table 3*; [*Crisp et al., 2013*; *Lindahl and Nyberg, 1972*; *Holmes et al., 2005*]) were used to calculate the relative rate difference over a given interval of the long-term temperature record and to quantify the offset from the reference temperature of 10°C, thus estimating the thermal age in years@10°C for each sample (*Figure 1C*). It is clear that Northern Hemisphere samples are thermally younger than their chronological age (e.g. Ellesmere Island is ~0.02 Ma@10°C), while the age of the eggshell samples considered here increases, e.g. the 3.8 Ma sample from Laetoli and the 1.34 Ma Olduvai samples are estimated to have thermal ages of ~16 Ma@10°C (*Appendix 1—table 4*; *Figure 1—source data 1*). The difference in chronological age between our two oldest sites is therefore minimised by the effect of temperature, which is dampened in Laetoli due to the greater altitude relative to Olduvai.

## Eggshell contains proteins (struthiocalcin-1 and -2) that bind very strongly to the calcite surface: good candidates for long-term survival?

This sample set, spanning the last ~16 Ma@10°C (*Table 1*), was chosen in order to explore patterns of diagenesis and protein survival using a well-established experimental approach that can isolate the intracrystalline fraction of proteins enclosed in biominerals, including ostrich eggshell (bleaching; *Crisp et al., 2013*). The intracrystalline fraction of avian eggshell typically contains C-lectins; in ostrich these are struthiocalcin 1 & 2 (SCA-1 & SCA-2) (*Mann and Siedler, 2004*). The eggshell proteins were characterised in terms of their amino acid composition across microstructural layers (Appendix 2, sections A and B) and the main proteins sequenced and identified (Appendix 2, section C) in modern eggshell, revealing uniform composition across the eggshell layers. Therefore, samples of the archaeological and paleontological eggshell, usually recovered in a fragmentary state, can be considered to be representative of the overall proteome.

The crystallography of SCA-1 (*Ruiz-Arellano et al., 2015*) reveals a similar overall structure to ovocleidin-17 (OC-17), which has previously been proposed to play a catalytic role in the calcification of chicken eggshell via the positively charged cluster of arginine residues interacting with the carbonates on the (10.4) calcite surface (*Freeman et al., 2010*). OC-17 is, however, absent in ostrich; instead, SCA-1 and 2 are negatively charged (*Table 2*) and thus likely to bind to calcium ions.

A molecular dynamics (MD) study of the binding of whole SCA molecules at the mineral surface allowed the strongest binding regions of SCA-1 and SCA-2 to be identified, two for each of the two proteins (Appendix 3). In MD simulations the four peptide sequences that cover the binding regions (*Table 2*) were moved close to the (10.4) calcite surface from aqueous (bulk) solution to determine their respective binding energies (Appendix 3). All four peptides showed negative binding energies,

**Table 2.** Binding of proteins to the (10.4) calcite surface. The binding energies calculated as (a) mean for the full protein (by minimization, see Appendix 3); (b) for four individual domains within the proteins (by molecular dynamics [ovocleidin]; by minimization [struthiocalcin]); (c) for the four domains treated as peptides (by molecular dynamics, see Appendix 3).

| | OC-17 | SCA-1 | SCA-2 |
|---|---|---|---|
| Charge on the protein | +7 (balanced by Cl⁻) | −11 (balanced by Na⁺, Ca²⁺) | −10 (balanced byCa²⁺) |
| Binding energy (mean): kJ mol⁻¹ | | −197 ± 22 | −142 ± 33 |
| Binding energy (domains): kJ mol⁻¹ | −422 ± 43 | −423 ± 42 (YHHGEEEEDVWI) <br> −611 ± 44 (YSALDDDDYPKG) | −255 ± 72 (SDSEEEAGEEVW) <br> −231 ± 68 (ASIHSEEEHQAIV) |
| Binding energy (peptides only) kJ mol⁻¹ | | −142 ± 19 (YHHGEEEEDVWI) <br> −219 ± 24 (YSALDDDDYPKG) | −131 ± 32 (SDSEEEAGEEVW) <br> −122 ± 41 (ASIHSEEEHQAIV) |
| Water molecules displaced | 21.3 | 20.2 | 23.1 |
| Estimated binding free energy: kJ mol⁻¹ | −188 ± 37 | −159 ± 24 | −99 ± 39 |
| Residence times for water (ps) [average surface bound water molecule = 120 ps] | | 130 ± 3 (YHHGEEEEDVWI) <br> 135 ± 3 (YSALDDDDYPKG) | 124 ± 4 (SDSEEEAGEEVW) <br> 123 ± 5 (ASIHSEEEHQAIV) |

indicating it was energetically favourable for them to bind to the calcite surface, rather than to remain in solution. SCA-1 bound more strongly than SCA-2 and the binding energies for all four peptides had the same relative order as in the simulations with full proteins. This indicates that the peptides operate as effective proxies for the binding of SCA. The differences between bindings of the different peptides are probably due to the individual amino acids and the primary structure of the peptide enabling favoured binding configurations.

When a molecule binds at the surface there will also be changes in entropy - an entropy loss for the molecule as it becomes bound and an entropy gain as water molecules on the surface are displaced. Given only one molecule binds, compared to the displacement of multiple water molecules, this will be an entropically favourable process. We have previously estimated the entropy associated with the water molecules and use this as a correction to the internal energy to estimate the free energy of binding (*Freeman and Harding, 2014*). These estimated free energy values (including the influence of water displacement) demonstrate the same trends as the configurational energy, since the number of water molecules displaced in all cases is similar.

The structure of the water close to the interface is also more ordered than bulk water and has lower energy. Thus, when hydrolysis of the bound protein or peptide occurs, it must react with the stabilized water at the interface, not the water in the bulk. This will raise the barrier to hydrolysis and thus promote the survival of the sequence (see also the discussion below). We would therefore expect that the stronger the peptide binding, the more likely the sequence is to survive in the fossil record, as it is best stabilized by its interactions with the mineral surface and must react with stabilized water. The MD simulations thus predict that the YSALDDDDYPKG sequence, with the lowest binding energy (*Table 2*) will survive the longest.

### Tracking protein breakdown in fossil ostrich eggshell: a multi-analytical approach to validate ancient sequences

For fossil eggshell samples the extent of degradation, quantified by chiral amino acid analysis (AAR), shows that both hydrolysis and racemization increase with time, and that racemic equilibrium is reached in samples older than 1 Ma (~3.6 Ma@10°C; *Figure 2*; and *Appendix 4—table 1* and *2*). As degradation proceeds, the complexity of the proteome decreases, until only SCA-1 and SCA-2 are detected by LC-MS/MS in the oldest samples (*Supplementary file 1*). These two proteins are extremely well preserved in samples up to 150,000 years old, but by 3.8 Ma few peptides are recovered (*Figure 3*, *Figure 3—figure supplement 1* and *2*). A total of 22 peptide sequences was recovered from SCA-1 & SCA-2 in samples from Laetoli (Appendix 5, section A; *Supplementary file 2*), consistent with the idea that dehydration, in addition to mineral binding, may also play a role in retarding degradation of non-binding peptides (*Collins and Riley, 2000*). However, 80% of the spectra, consistently identified in ten independent LC-MS/MS analyses of three ostrich eggshell samples

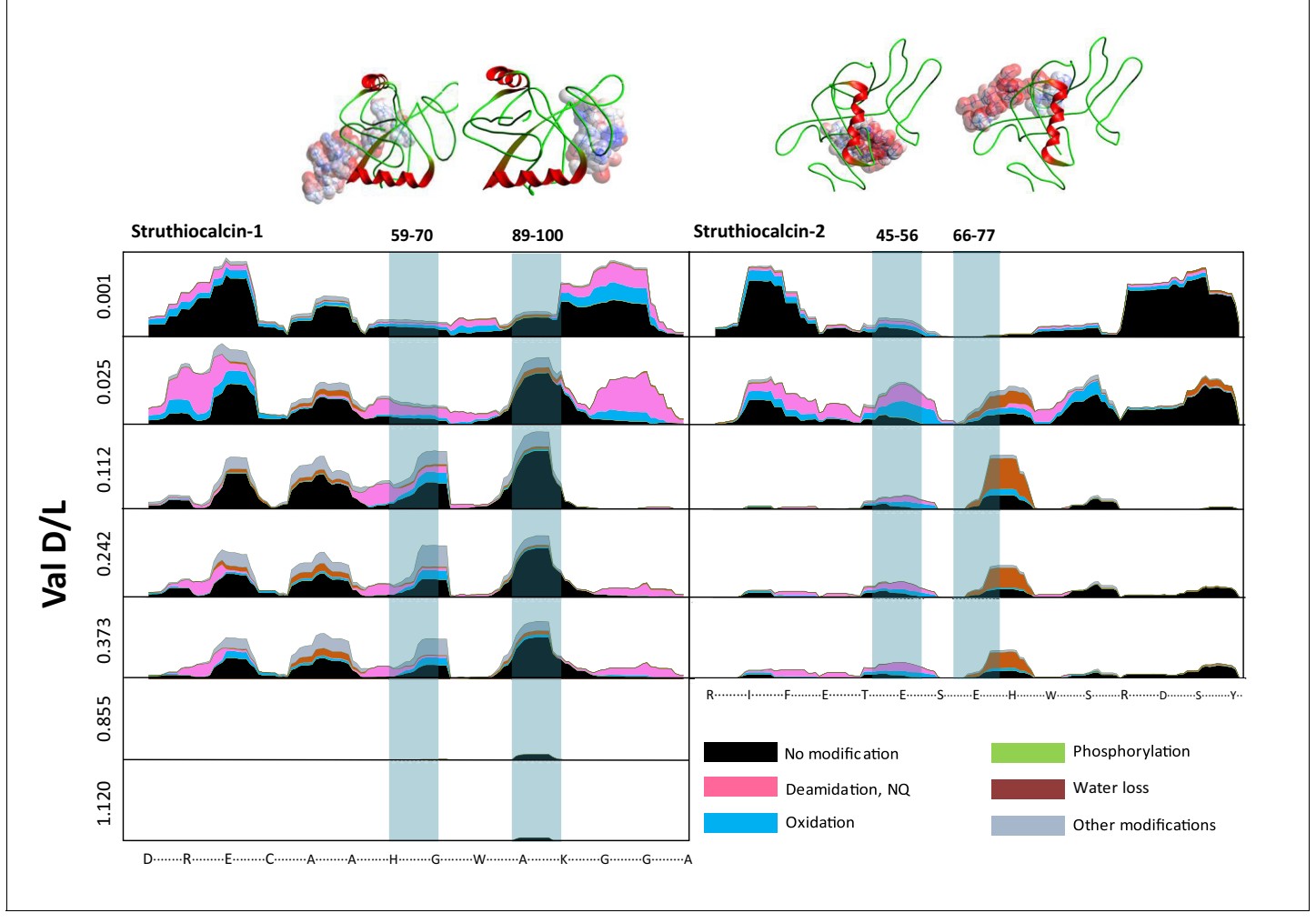

**Figure 3.** Survival of struthiocalcin 1 and struthiocalcin 2 peptides. Over time (and increasing THAA Val D/L values) the spectral count decreases as degradation progresses. Blue bars highlight the peptides investigated computationally (represented by the filled atoms in the models). Highly degraded samples (Val D/L ~0.8–1.1) preserve the DDDD-containing peptide. The full time series is shown for SCA-1 in *Figure 3—figure supplement 1* and for SCA-2 in *Figure 3—figure supplement 2*.

The following figure supplements are available for figure 3:

**Figure supplement 1.** Frequency of identified spectra of SCA-1 in bleached OES (fossils) and purified proteins (kinetics).

**Figure supplement 2.** Frequency of identified spectra of SCA-2 in bleached OES (fossils) and purified proteins (kinetics).

from Laetoli were assigned to charged species that contained the four Asp residues found in the peptide YSALDDDDYPKG. The survival of this Asp-rich peptide region is not limited to the samples from Laetoli; the eggshells from Olduvai (~16 Ma@10°C) and Wonderwerk (~3.6 Ma@10°C) also show that this region of SCA-1 is preferentially preserved.

This peptide does not survive in the absence of the mineral, as shown by the artificial degradation experiments we conducted on purified eggshell proteins heated at 140°C in water (*Figure 2— figure supplement 2*, Appendix 4, sections B and C). Indeed, the same region of the protein was too flexible to be determined when the crystallographic structure of pure SCA-1 was solved (*Ruiz-Arellano et al., 2015*). The patterns of degradation of the same proteins heated in water *vs* after demineralisation of the eggshell mineral also differ significantly from each other. It is noteworthy that the hydropathicity calculated for the surviving peptides decreases in eggshell mineral and

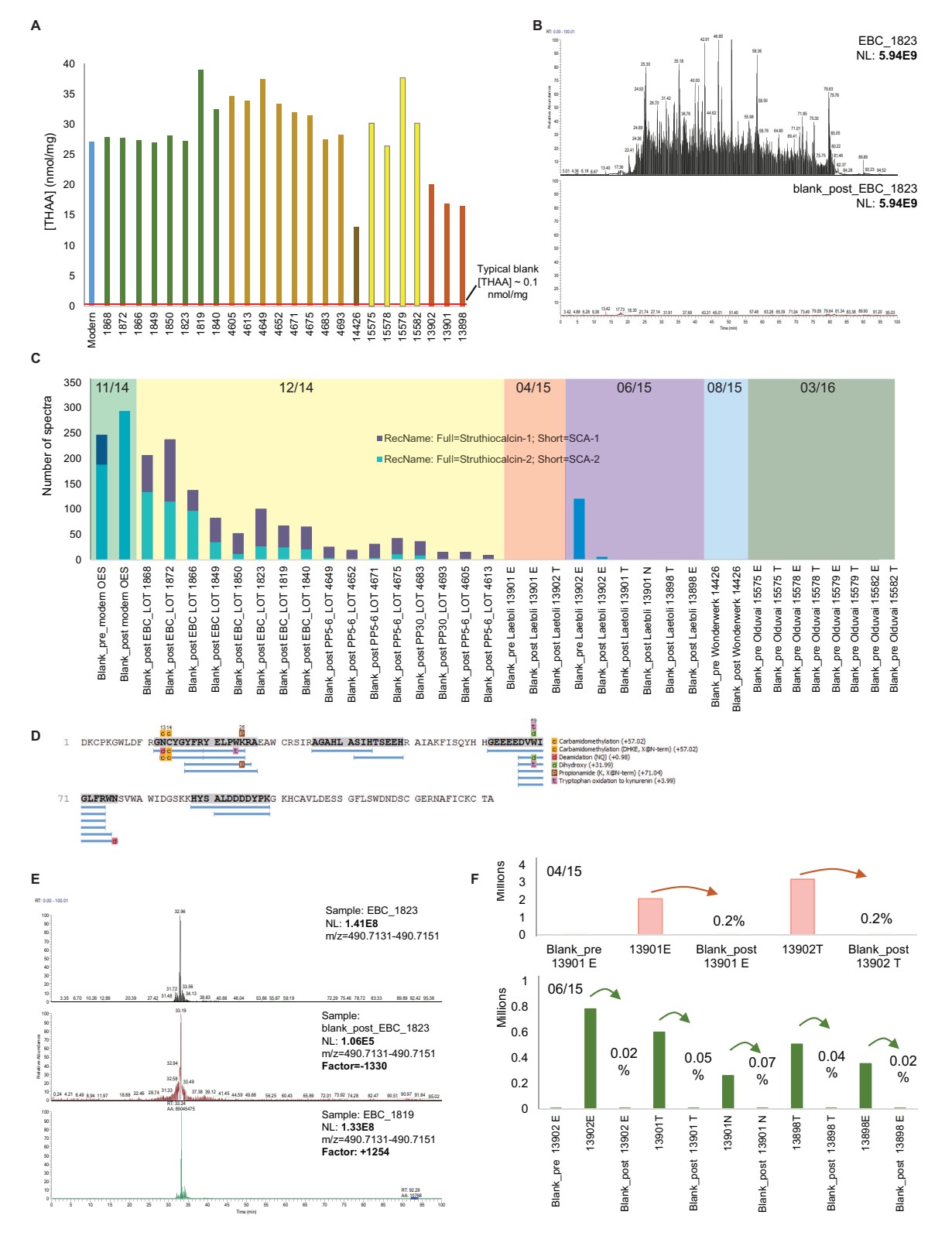

**Figure 4.** Authenticity of the ancient sequences. Amino acid analyses (**A**): Total concentrations in all eggshell samples (sum of Asx, Glx, Gly, Ala, Val and Ile). Carry-over: (**B**) Total ion chromatogram for one eggshell sample (EBC_1823) and the blank analysed immediately after (blank_post_EBC1823). (**C**) Spectral abundance of SCA-1 and SCA-2 in LC-MS/MS blanks. (**D**) SCA-1 coverage in the blank analysed after a Pinnacle Point eggshell sample PP_4652. Note 'DDDD-' and 'EEEED-' peptides and Asn deamidation. (**E**) Extracted ion chromatogram for LDDDDYPK in EBC_1823,

*Figure 4 continued on next page*

*Figure 4 continued*

blank_post_EBC1823 and EBC_1819. (**F**) Absolute and relative total abundance of 'DDDD' peptides in Laetoli samples/blanks. Signal reduction is at least 100-fold (more often 1000- or 10,000-fold). Independent replication and manual *de novo* sequencing of the peptides from Laetoli (Appendix 5, section A; *Supplementary file 2*), consistency of diagenesis-induced modifications (Appendix 5, section D; *Supplementary file 3*) and volatile organic compound analyses (Appendix 5, section E) were also used to validate the results obtained.

increases in water (*Figure 2* and *Figure 2—figure supplement 2*), consistent with the hypothesis that mineral binding plays a crucial role in the survival of selected peptide sequences.

The authenticity of the peptide sequences recovered in the oldest samples was thoroughly assessed (Appendix 5). The amino acid concentration was analysed in all bleached eggshell samples and controls (procedural blanks); concentrations in the blanks were negligible, while the samples retain the original organic fraction within the intracrystalline environment (*Figure 4A*). In addition, the presence of volatile organic compounds in 2.7 Ma ostrich eggshell demonstrates the stability of ostrich eggshell as a closed system (Appendix 5, Section E). Ratite eggshell has previously proven to be an excellent source of ancient DNA (*Oskam et al., 2010*) but, unsurprisingly, NGS sequencing failed to recover avian DNA from the Laetoli eggshell we tested (Appendix 5, Section F). Water blanks were injected between each LC-MS/MS eggshell sample analysis to assess carry-over (*Figure 4B–F*). Despite low levels of SCA-1 being occasionally detected in some of the blanks (*Figure 4D*), the effective carry-over from sample to sample can be estimated to be below 0.01%. We also stress that each batch of fossil eggshell was analysed separately in time (*Figure 4C*) and that therefore carry-over between younger and older eggshell samples is impossible. Finally, independent analyses of the results in a second laboratory (Copenhagen) also demonstrated the replicability of our results (Appendix 5, Section A). All peptides and proteins detected in this study presented damage patterns (i.e. diagenesis-induced modifications, such as deamidation, oxidation) that are entirely consistent with the age of the samples (Appendix 5, Section D; *Supplementary file 3*).

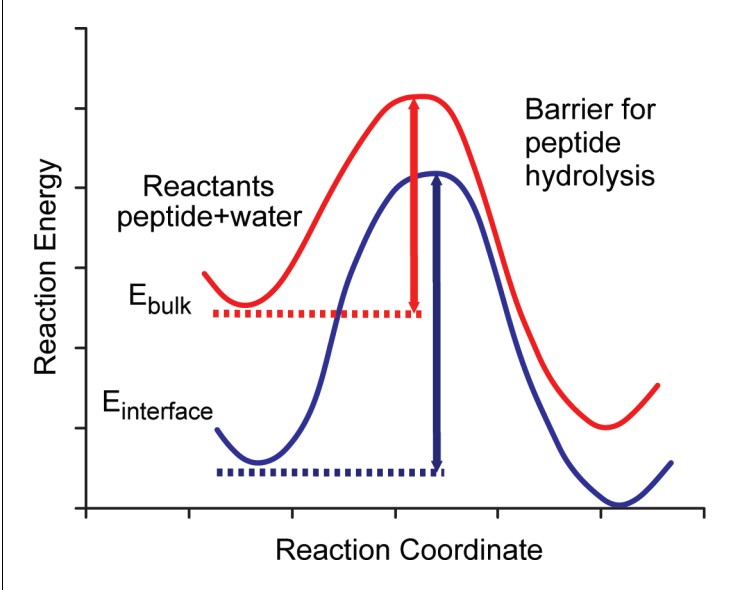

**Figure 5.** Schematic diagram of energy barriers for peptide hydrolysis. A pictorial representation of the energy barriers associated with the lysis of the peptide. The process in bulk water is depicted in red and the process at the surface is depicted in blue. The surface process shows a larger barrier due to the stabilization of the reactants at the surface.

## Discussion

### Surface stabilization is the mechanism for long-term survival of biomineralizing peptides

The breakdown of the proteins and peptides should primarily occur via hydrolysis, involving water and proteins or peptides as reactants. The rate-determining step is the attack of a water molecule (or molecules, see *Pan et al., 2011* for an extended discussion). We schematically map out the pathway in *Figure 5* (red line) where the reaction coordinate denotes approach of water to the peptide and their subsequent reaction. The process requires energy (heat) to be given to the system in order to overcome the energy barrier. In hot environments, such as Tanzania, the high ambient heat means that many interactions have sufficient energy to overcome this barrier, yet our experimental findings demonstrate that some peptides survive.

We argue that the mechanism allowing the survival of the ancient sequences over ~4 Ma (~16 Ma@10°C) at equatorial sites is the stabilization of optimally configured peptides and associated water molecules by surface binding at this interface. The low, negative free energy of binding (*Table 2*) of the amino acid residues means that they will readily bind to the calcite surface and remain bound indefinitely and this binding stabilizes the peptides by lowering their configurational energy (*Table 2*). Thus, both the position of the ground state and the top of the barrier will be lowered with respect to the situation when the peptide is in bulk water (*Figure 5*). The binding of the peptide also forces the hydrolysis reaction to take place with the stabilized water close to the calcite surface.

Furthermore, the presence of the calcite surface significantly stabilizes the water molecules surrounding the peptide. Estimates of the residence times (*Table 2*) and diffusion values of water molecules trapped between the protein and mineral surface indicate that these water molecules have greater residence times and lower diffusion rates than water molecules on the surface with no protein present. This large stabilization of water molecules selectively lowers the ground state energy of the reactants (protein or peptide plus water) at the interface with respect to the bulk. Thus the energy barrier will be significantly larger for the bound protein or peptide than for the unbound one. Our surface molecules would therefore need more energy in the system (i.e. a higher temperature) to overcome the augmented barrier. The net effect of the binding of the protein or peptide is therefore to retard hydrolysis and prolong peptide sequence survival, albeit of a select (mineral-binding) region of the protein.

Translating this concept to real samples in geological settings, the burial temperature in Tanzania, which may be high enough to allow rapid hydrolytic breakdown of most proteins, would not be enough to hydrolyse mineral bound peptides over corresponding timescales because of their own stabilization but particularly because they are surrounded by the stabilized water. This is effectively equivalent to a localised 'cooling' effect: the water molecules at the calcite surface would therefore be expected to operate as if they were 'cooler' in terms of reactivity and rates of peptide bond hydrolysis.

### Conclusions: biominerals are a source of ancient protein sequences, preserved over geological timescales

The survival of 3.8 Ma old peptide sequences in equatorial Africa corresponds to an estimated thermal age of ~16 Ma@10°C, two orders of magnitude beyond the oldest recovered DNA (*Figure 1*). We explain this exceptional preservation in terms of surface stabilization of both the peptide and water molecules involved in the hydrolytic breakdown of the peptide. Our discovery identifies mineral-binding proteins as the most likely source of ancient biomolecular sequences in the fossil record.

In this study, we also set out parameters for the authentication of ancient sequences: the combination of the consistency in patterns of protein degradation and survival of particular peptide regions, independent replication of the results and an in-depth analysis of analytical blanks provide overwhelming evidence for the endogeneity and integrity of the peptides recovered. We suggest that all ancient proteomics studies undertake a similar approach to verify the authenticity of the sequences reported.

We anticipate that this study will open up new avenues in palaeontology and palaeoanthropology, for the first time enabling direct comparison between morphological and molecular records of fossils in deep time. Furthermore, the selective preservation of domains associated with biomineralization offers a novel strategy for uncovering functional regions governing mineral formation.

## Materials and methods

### Summary

Ostrich eggshell samples were ground and bleached for 72 hr (NaOCl 12% wt/vol) and rinsed thoroughly before demineralization. Amino acid and mass spectrometry analysis of ancient proteins was conducted using published techniques (*Buckley et al., 2009*; *Crisp et al., 2013*). Modifications include the use of trypsin and elastase as digestion enzymes for separate preparations (*Welker et al., 2015*). Liquid chromatography tandem mass spectrometry (LC-MS/MS) analyses were performed on Thermo Scientific Orbitrap platforms. Resulting spectra were searched against the Struthioniformes genomes using PEAKS (version 7.5 [*Ma et al., 2003*]). For PEAKS, FDR rate was set at 0.5%, with proteins accepted with $-10$lgp scores $\geq 40$ and ALC (%) $\geq 80$. A combination of minimization and conventional MD using the DL_Poly Classic code was used to explore possible protein–calcite binding geometries (*Freeman et al., 2011*).

### Materials

#### Modern ostrich eggshell (OES)

Modern OES was purchased from Oslinc, an ostrich farm based in Boston, Lincolnshire, UK (www.oslinc.co.uk). The complete shell was less than 1 year old; for the purposes of reproducibility all analyses were performed using the same eggshell. Modern OES was used for the investigation of the amino acid composition of the microstructural layers as well as high-temperature studies on purified proteins (kinetic experiments).

#### Archaeological and palaeontological OES

The details of the archaeological and palaeontological samples included in this study are detailed in *Tables 3–7*. LOT numbers and individual NEaar (sample) numbers were attributed at the NEaar laboratory, University of York.

### Elands Bay Cave

Elands Bay Cave (EBC) is located on the present coastline about 200 km north of Cape Town (South Africa). Human occupation occurred repeatedly since the terminal Pleistocene. Ostrich eggshell is present throughout the sequence. The site's chronology has been firmly established through multiple radiocarbon dates; the samples analysed in this study can each be assigned to an age range on the basis of dates obtained on each layer and/or bracketing the OES (*Parkington, 1980*; *Stowe and Sealy, 2016*).

### Pinnacle Point

The caves at Pinnacle Point (PP) have been in the spotlight of archaeological research for the past few years, and they have yielded extraordinary evidence for early modern human behaviour as well as detailed palaeoclimatic information (*Karkanas et al., 2015*; *Bar-Matthews et al., 2010*; *Brown et al., 2009*; *Marean, 2010*; *Marean et al., 2007*). The OES fragments analysed in this study come from two sites in the PP complex, PP 5–6 and PP 30, and were selected from excavations done up to 2010. PP 5–6 is a well-dated sequence, spanning ca. 50–90 ka. PP 30 is a hyena den (~150 ka BP) and the OES material reflects a single depositional episode.

### Wonderwerk Cave

Wonderwerk Cave (WW) is located in the arid interior of South Africa, near the southern border of the Kalahari Desert. The site has yielded a unique ca. 2 million years long archaeological sequence (*Horwitz and Chazan, 2015*; *Berna et al., 2012*). Stratum 10, from which the OES samples analysed here are derived, bears the earliest evidence of intentional use of fire during the Acheulean,

**Table 3.** Summary of samples from Elands Bay Cave, South Africa. The stratigraphic layers have been independently dated by radio-carbon. Unpublished uncalibrated dates provided by J. Parkington. Date calibration was performed with OxCal v.4.2 (**Ramsey, 2009**. Calibration curves: IntCal13 for dates obtained on charcoal; Marine13 for dates obtained on shells/crayfish, DeltaR = 93 ± 28 [**Dewar et al., 2012**]). Age estimates for undated layers based on estimating the median (mid-point) of two dates obtained on layers bracketing the layer with OES samples.

| LOT | NEaar | Layer | Age (cal BP) 95.4% | Material used for $^{14}$C dating/notes |
|-----|-------|-------|---------------------|------------------------------------------|
| 1868 | 6887 | Kaunda | <323 (estimate) | Layer above dates on layer NKOM |
| 1872 | 6888 | George Best | 322–1008 | Layer between dates on layers NKOM and EDDI |
| 1866 | 6889 | D. Lamour | 906–2282 | Layer between dates on layers EDDI and LARM |
| 1849 | 6891 | Maroon Robson | 8773 ± 125 | Charcoal |
| 1850 | 6893 | Nero | 8748–10096 | Layer between dates on layers Maroon Robson / Burnt Robeson |
| 1823 | 6896 | Crayfish | 11545 ± 441 | Crayfish |
| 1819 | 6899 | Smoke | 12589 ± 104 | Charcoal |
| 1840 | 6907 | OBS 1 | 15208–15940 | Layer between dates on layers Smoke and SOSE |

constrained to the Jaramillo subchron (1.07–0.99 Ma) based on a combination of paleomagnetic and cosmogenic burial age dating (**Horwitz and Chazan, 2015**; **Berna et al., 2012**). The OES fragments from the cave have been used as an effective proxy for refining palaeoclimatic and environmental reconstructions, especially for the early-mid Pleistocene and Holocene levels (**Ecker et al., 2015**; **Lee-Thorp and Ecker, 2015**).

## Olduvai

Olduvai Gorge (Tanzania) contains an extensive record of the past two million years of human evolution. The eggshell samples analyzed in the present study were found in situ during the excavation of the BK (Bell Korongo) site located in uppermost Bed II. The site is exceptional by the large amount of ostrich eggshell fragments that were found throughout all its stratigraphic sequence. A volcanic tuff just underlying BK was recently dated to ~1.34 Ma (**Domínguez-Rodrigo et al., 2013**). The samples analyzed were found in Level 4, which contains a wealth of fossil bones and associated stone tools. This level has been interpreted as a central-place where the butchery of several animal carcasses (including megafaunal remains from *Sivatherium* and *Pelorovis*) was carried out (**Domínguez-Rodrigo et al., 2014**).

## Laetoli

Laetoli (Tanzania) is one of the most famous sites for palaeoanthropologists: it has yielded hominin and other animal remains (**Harrison, 2011a**, **2011b**) and the first unequivocal evidence for

**Table 4.** Sample details for sub-fossil OES analysed for LC-MS/MS; from Pinnacle Point, South Africa. Stratigraphic information and weighted mean OSL age estimates (ka) for PP 5–6 (**Karkanas et al., 2015**) and PP 30 (**Rector and Reed, 2010**).

| Site | LOT | NEaar | Archaeological sample information | Stratigraphic aggregate | Age (ka) |
|------|-----|-------|----------------------------------|-------------------------|----------|
| PP5-6 | 4613 | 7676 | Plotted Find 102627, Lot 3151 | RBSR | 51 ± 2 |
| PP5-6 | 4649 | 7283 | Plotted Find 165702, Lot 8038 | SGS | 64 ± 3 |
| PP5-6 | 4671 | 7316 | Specimen 273467, Lot 3255 | SADBS | 71 ± 3 |
| PP5-6 | 4605 | 7198 | Specimen 273489, Lot 3277 | SADBS | 71 ± 3 |
| PP5-6 | 4652 | 7286 | Plotted Find 178331, Lot 8172 | ALBS | 72 ± 3 |
| PP5-6 | 4675 | 7320 | Specimen 273514, Lot 7980 | LBSR | 81 ± 4 |
| PP 30 | 4683 | 7328 | Specimen 66008, Lot 1795 | Single horizon | ~151 |
| PP 30 | 4697 | 7342 | Specimen 65168, Lot 1750 | Single horizon | ~151 |

**Table 5.** Sample details for sub-fossil OES samples from Wonderwerk Cave, South Africa. Ages based on cosmogenic isotope burial dating and magnetostratigraphy, from *Matmon et al. (2012)*.

| LOT | NEaar | Stratum | Independent age (Ma) |
|-----|-------|---------|----------------------|
| 14426 | 10581 | ME46, SPF#4390, Exc. 1, stratum 10, square Q33, depth 15–20 cm | 1.07–0.99 |

bipedalism thanks to the footprints of *Australopithecus afarensis* preserved in Pliocene volcanic ash, discovered by Mary Leakey in 1976 (*Leakey and Hay, 1979*).

The eggshell at Laetoli are surface finds, but visual examinations show no evidence of rolling, transportation or weathering (having been exposed on the surface for only a very short period of time after having eroded out of the sediment). As a consequence, there is no likelihood of long-distance transport. The location and preservation of the fossils, the absence of significant spatial displacement of surface finds, the short stratigraphic sections at each collecting spot, and the identification of the fossil-bearing horizons in each of those sections, allow the fossils to be placed quite precisely in their original stratigraphic context. The age and stratigraphy given for each of the samples can be assigned with a high degree of confidence. There are no lava flows in stratigraphic proximity or direct superposition to the stratigraphic units from which the Lower Laetolil and Upper Ndolanya specimens were recovered. Given that more than 40 m of consolidated sediment, and a time difference of 1.5 million years, separate the overlying lava flow (the Ogol Lavas) from the stratum from which the Upper Laetolil fossils were obtained, and that the intervening fossil-bearing beds show no geological evidence of having been impacted by heating, we do not believe that the samples have been exposed to additional heating that would have made them thermally older than we predict (*Table 7*).

## Methods

### Bleaching pre-treatment

All analyses reported in this study were conducted on bleached ostrich eggshell (OES) in order to isolate the functionally intra-crystalline proteins. Based on the results of *Crisp et al. (2013)*, powdered eggshell was submerged in NaOCl (12% w/v) for a minimum of 72 hr.

### Chiral amino acid (AAR) analyses

Sample preparation for chiral amino acid analyses (total hydrolysable and free amino acids fractions: THAA and FAA) was carried out following the method of *Crisp et al. (2013)*. Separation of the chiral forms (D and L) of multiple amino acids was performed by RP-HPLC with fluorescence detection using a modified method of *Kaufman and Manley (1998)*. The amino acids reported here are among those detected routinely with good chromatographic resolution in OES: Asx and Glx (Asp +Asn and Glu +Gln due to irreversible deamidation during sample preparation), alanine (Ala), valine (Val) and isoleucine (Ile). Serine (Ser) is not reported as its decomposition patterns are complicated by decomposition and a reversal in D/L values, therefore its utility decreases for older samples (*Vallentyne, 1964*; *Kimber et al., 1986*).

**Table 6.** Sample details for fossil OES samples from Olduvai, Tanzania.

| LOT | NEaar | Locality/Stratum | Independent age (Ma) |
|-----|-------|------------------|----------------------|
| 15575 | 10955 | Sample BK09-3150 | 1.338 ± 0.024 |
| 15578 | 10958 | Sample BK10-5309 | 1.338 ± 0.024 |
| 15579 | 10959 | Sample BK09-2627 | 1.338 ± 0.024 |
| 15582 | 10962 | Sample BK09-2706 | 1.338 ± 0.024 |

**Table 7.** Sample details for fossil OES samples from Laetoli, Tanzania. Ages of the strata and locali-ties ($^{40}$Ar/$^{39}$Ar) from **Deino (2011)**. LOT 13901 is attributed to *Struthio camelus*. LOTs 13902 and 13898 are attributed to *Struthio kakesiensis* (**Harrison and Msuya, 2005**).

| LOT | NEaar | Locality/Stratum | Independent age (Ma) |
|------|-------|------------------|----------------------|
| 13901 | 10574 | Loc 15, Upper Ndolanya Beds | ~2.66 |
| 13902 | 10573 | Loc 10 West, Upper Laetolil Beds | ~3.8–3.85 |
| 13898 | 10575 | Kakesio 1−6, Lower Laetolil Beds | ~3.85 -> 4.3 |

## Proteomics

### High-temperature experiments: purified proteins from modern OES

Bleached modern OES powders were demineralized in cold dilute acetic acid (10% v/v, 4°C, over-night). The solution was centrifuged at 4500 RPM for 1 hr at 4°C, ultrafiltered (Amicon ultra-filters, 10 kDa) and rinsed repeatedly with ultrapure water. The concentrated proteins were lyophilized and resuspended in ultrapure water. 125 µL of suspension was transferred to four individual sterile hydro-lysis vials, each sealed with a clean teflon cap and heated at 140°C for 2, 8, 24 and 120 hr respec-tively. Alkylation / reduction of disulphide bonds was carried out using dithiothreitol (60°C, 60 min; Sigma Aldrich, St Louis, MO) and iodoacetamide (room temperature, 45 min; Sigma Aldrich). The solutions were then dried down in a centrifugal evaporator to be analysed directly by LC-MS/MS.

### Protein extraction: archaeological OES (York)

The average sample size for proteomics was 35 mg of OES powder. Two separate preparations were carried out on two subsamples (~17 mg each), for digestion with trypsin ('T') and elastase ('E'). All subsamples were demineralized in cold 0.6 M HCl and the solution neutralized, lyophilized and resuspended in ammonium bicarbonate or Tris-HCl buffer containing the RapiGest SF surfactant (1 mg/mL; Waters Ltd, Hertfordshire, UK), for 'T' and 'E' subsamples, respectively. Following reduction and alkylation of disulphide bonds with DTT and IAA according to the usual protocols, digestion was carried out overnight at 37°C by adding: 4 µL trypsin (0.5 µg/µL; Promega, 2800 Woods Hollow Road Madison, WI 53,711 USA) for 'T' subsamples or 4 µL elastase (1 µg/µL; Worthington, Lake-wood, NJ, USA) for 'E' subsamples.

Digestion was stopped by adding trifluoroacetic acid (TFA) to a final concentration of ~0.1% (v/v) and RapiGest™ precipitated by incubating in an acidic environment at 37°C for 30 min. Samples were centrifuged on a bench-top centrifuge (13000 RPM, 30 min) and purified using C$_{18}$ solid-phase extraction (Pierce zip-tip; Thermo-Fisher) according to the manufacturer's instructions. Eluted pepti-des were evaporated to dryness using a centrifugal evaporator before LC-MS/MS analyses.

### Protein extraction: Laetoli OES (Copenhagen)

The Laetoli eggshell samples (LOT 13901r, 13902r, 13898r), powdered and pre-bleached in York, were sent to the University of Copenhagen for replication. A negative control sample, prepared exactly like the ancient ones except for the initial addition of eggshell powder, was processed and analysed together with the ancient samples, following the same procedure. All samples, including the negative extraction control, were processed in laboratories regularly used for ancient DNA extraction, implementing all the measures necessary to avoid potential contamination from modern biomolecules. All surfaces were UV irradiated overnight, and repeatedly cleaned with bleach and ethanol. In addition, facemasks, nitrile gloves, hairnets and body suits were worn continuously by operators.

An aliquot of 58 mg, 48 mg and 64 mg was weighed from samples 13902r, 13901r and 13898r respectively, and placed in 1.5 mL Protein Lo Bind Tubes (Eppendorf). Subsequently, they were sus-pended in 1 mL 0.5 M EDTA pH 8.00, mechanically shaken for approximately one minute and incu-bated overnight under rotation at room temperature. The following day, after centrifugation at 17,000 g for 10 min, the EDTA supernatant was removed and stored in a −18°C freezer. The demin-eralisation step with 1 mL 0.5 M EDTA was repeated one more time. The third day, after removal of EDTA supernatant, all demineralised pellets were re-suspended with 100 µL of 0.1 M Tris pH 8.00,

mechanically shaken for approximately one minute, and precipitated by centrifugation at 17,000 *g* for 10 min. The wash step with 100 µL of 0.1 M Tris was repeated two more times.

The samples were then further processed in a guanidinium hydrochloride lysis buffer solution following published methods (*Kulak et al., 2014*; *Jersie-Christensen et al., 2016*), without sonication or equivalent steps. Samples were instead mechanically shaken for approximately one minute and a micro-pestle (Eppendorf) was used to manually disrupt the pellet. Ancient samples and negative control were initially diluted to 1:3 in dilution buffer (*Kulak et al., 2014*), 0.5 µg of rLysC (Promega) were added, and the solution was digested at 37°C, with mechanical shaking at 900 rpm, for two hours. Samples were then further diluted 1:3 in dilution buffer, 0.5 µg of mass spectrometry grade trypsin (Promega) were added, and the solution was digested overnight at 37°C, with mechanical shaking at 900 rpm. On the following day, samples were acidified, using 10% trifluoroacetic acid (TFA) in ultrapure water, to reach pH < 2.00, and then centrifuged at 17,000 *g* for 1 hr. The resulting peptide mixtures in the supernatant fraction were then concentrated using in house created $C_{18}$ solid phase extraction stage tips as described by *Cappellini et al. (2012)*.

## LC-MS/MS analysis

### Oxford TDI

Subsamples digested with trypsin and elastase were combined in a single LC-MS/MS run, with the following exceptions:

- Subsamples E and T for all Olduvai and Laetoli OES
- Subsample LOT 13901N: digestion step was not performed
- Purified proteins heated at high temperature in water (kinetics): digestion step was not performed

Elastase and trypsin-digested samples were analysed by LC-MS/MS as described before (*Fischer and Kessler, 2015*). Briefly, peptides were separated on a PepMAP $C_{18}$ column (75 µm × 500 mm, 2 µm particle size, Thermo) using a Dionex Ultimate 3000 UPLC at 250 nL/min and Acetonitrile gradient from 2–35% in 5% DMSO/0.1% formic acid. Peptides were detected with a Q-Exactive mass spectrometer (Thermo) at a resolution of 70000 @ *m/z* 200 and an ion target value of 3e6 between *m/z* 380 and 1800. Up to 15 precursors were selected for HCD fragmentation at a resolution of 17,500 with an ion target of 1e5 and a maximal injection time of 128 ms. Normalized collision energy was fixed at 28% and the isolation windows was 1.6 *m/z* units.

### Copenhagen

The samples were separated on a 50 cm PicoFrit column (75 µm inner diameter) in-house packed with 1.9 µm $C_{18}$ beads (Reprosil-AQ Pur, Dr. Maisch) on an EASY-nLC 1000 system connected to a Q-Exactive HF (Thermo Scientific, Bremen, Germany). The peptides were separated with a gradient going from 2% to 25% buffer B in 110 min followed by a 25 min step to 40%. After the gradient the column was washed by going to 60% in 5 min, held for 5 min and re-equilibrated back to 2% for 15 min, resulting in a final acquisition of 165 min. Buffers contained 0.1% TFA dissolved in either 80% acetonitrile for buffer B, or milli-Q water for buffer A. The flow rate was 200 nL/min throughout the gradient and wash.

The Q-Exactive HF was operated in data-dependent top 10 mode. Full scan mass spectra were recorded at a resolution of 120,000 at *m/z* 200 over the *m/z* range 300–1750 with a target value of 3e6 and a maximum injection time of 20 ms. HCD-generated product ions were recorded with a maximum ion injection time set to 108 ms through a target value set to 2e5 and recorded at a resolution of 60,000 with a fixed first mass set to *m/z* 100. Normalized collision energy was 28%. The isolation window was set at 1.3 *m/z* units and the dynamic exclusion to 30 s.

## Identification of peptides and proteins

Product ion spectra were analysed using the software PEAKS Studio (v. 7.0, Bioinformatics Solutions Inc. (BSI) [*Ma et al., 2003*]). Mascot generic format (mgf) files were searched against a reference database containing the genomes of all Struthioniformes and common contaminants (40566 entries), assuming no digestion enzyme and with fragment ion mass tolerance of 0.050 Da and a parent ion tolerance of 5.0 ppm. Results obtained by SPIDER searches (i.e. including all modifications) were used for the investigation of protein survival in OES using the following threshold values for

acceptance of high-quality peptides: false discovery rate (FDR) threshold 0.5%, protein scores $-10lgP \geq 40$, *de novo* sequences scores (ALC% ) $\geq 80$.

## Volatiles

On crushing or demineralisation of the subfossil OES, a strong odour was emitted from some samples, so analysis of these volatiles was attempted using gas chromatography mass spectrometry (GC-MS). A sealed container was designed that allowed in-line crushing of a sample under $N_2$. Volatiles emitted during crushing of the shells were measured using thermal desorption (Unity, Markes International, Llantrisant, UK) coupled to gas chromatography with a high-resolution quadrupole time of flight mass spectrometer (7200B GC/Q-TOFMS, Agilent Technologies,Wilmington, DE, USA). Volatile organic compounds (VOCs) were flushed from the shell crusher onto the trap using high purity nitrogen at 100 mL min$^{-1}$ for 10 min. The trap was held at $-30°C$ during sampling, then ballistically heated to 250°C and held for 5 min to ensure complete desorption. The heating of the trap triggered the start of the GC run. A 5% phenyl-polysilphenylene-siloxane capillary column was used (50 m $\times$ 0.32 mm $\times$ 1 mm BPX5, SGE, Australia) to separate the VOCs. The oven was held at 40°C for 5 min, followed by a ramp rate of 10°C min$^{-1}$ to a final temperature of 230°C, which was held for 3 min. High purity helium gas was used as the mobile phase at a flow rate of 4.5 ml min$^{-1}$. The mass spectrometer was operated in an electron ionisation mode at 70 eV and the ion source was at 250°C. Spectra were collected between *m/z* 35 and 500 at an acquisition rate of 5 spectra s$^{-1}$. The mass spectra obtained were compared to the NIST MS database (NIST MS Search Program version 2.0) after background correction. A nitrogen blank, sampled through the shell crusher was used to determine the method background and identify unique VOCs emitted from the egg shells. Analysis was undertaken on a fragment of one of the subfossil OES from Laetoli (LOT 13901, ~2.7 Ma).

## Ancient DNA

Two DNA extractions using ~0.05 g of eggshell (sample Laetoli LOT 13902) were made following *Dabney et al. (2013)* and the extracts combined before the final elution in 25 μL TET.

## Data availability

The data discussed in the paper are archived in the following databases: the mass spectrometry proteomics datasets have been deposited to the ProteomeXchange Consortium via the PRIDE partner repository with the dataset identifier PXD003786; Illumina genetic data have been deposited in the NCBI Short Read Archive (SRA), BioProject ID PRJNA314978; computational modelling data can be found at DOI: 10.15131/shef.data.3491387 (this contains pdb files giving the initial configurations used for SCA-1, SCA-2 and the four peptide sequences and input files for DL_POLY that contain a complete specification of the forcefield used and other setting parameters for the simulations).

## Acknowledgements

The authors are grateful to Tom Gilbert, Jessica Hendy and Darrell Kaufman for useful discussion and support. Additional funding: work at the Centre for GeoGenetics was financed by the Danish National Research Foundation (DNRF94).

## Additional information

### Funding

| Funder | Grant reference number | Author |
|---|---|---|
| Engineering and Physical Sciences Research Council | EP/I001514/1 | Beatrice Demarchi<br>Teresa Roncal-Herrero<br>Colin L Freeman<br>Roland Kröger<br>John H Harding<br>Matthew J Collins |
| Natural Environment Research | NERC NE/G004625/1 | Beatrice Demarchi |

| | | |
|---|---|---|
| Council | | Molly K Crisp<br>Julia Lee-Thorp<br>Kirsty Penkman<br>Matthew J Collins |
| Arts and Humanities Research Council | AHRC AH/L006979 | Beatrice Demarchi<br>Matthew J Collins |
| European Research Council | PERG07-GA-2010-268429 | Beatrice Demarchi |
| European Research Council | SMILEY FP7-NMP-2012-SMALL-6-310637 | Teresa Roncal-Herrero<br>Roland Kröger |
| Engineering and Physical Sciences Research Council | EP/K000225/1 | Colin L Freeman<br>John H Harding |
| Engineering and Physical Sciences Research Council | EP/L000202 | Colin L Freeman<br>John H Harding |
| Novo Nordisk | NNF14CC0001 | Rosa Rakownikow Jersie-Christensen<br>Jesper V Olsen |
| Danish National Research Foundation | DNRF94 | James Haile |
| National Science Foundation | BCS-0524087 | Curtis W Marean |
| Hyde Family Foundations | | Curtis W Marean |
| John Templeton Foundation | | Curtis W Marean |
| Arizona State University | | Curtis W Marean |
| National Science Foundation | BCS-1138073 | Curtis W Marean |
| National Science Foundation | 547414 | Samantha Presslee<br>Ross DE MacPhee<br>Matthew J Collins |
| National Geographic Society | | Terry Harrison<br>Amandus Kwekason |
| Leakey Foundation | | Terry Harrison<br>Amandus Kwekason |
| National Science Foundation | BCS-9903434 | Terry Harrison<br>Amandus Kwekason |
| National Science Foundation | BCS-0309513 | Terry Harrison<br>Amandus Kwekason |
| National Science Foundation | BCS-0216683 | Terry Harrison<br>Amandus Kwekason |
| Tanzanian Department of Antiquities | Permit | Manuel Domínguez-Rodrigo |
| Social Sciences and Humanities Research Council of Canada | | Liora Kolska Horwitz<br>Michael Chazan |
| Leverhulme Trust | | Kirsty Penkman |
| Spanish Ministry of Economy and Competitivity | HAR2013-45246-C3-1-P | Manuel Domínguez-Rodrigo |

The funders had no role in study design, data collection and interpretation, or the decision to submit the work for publication.

## Author contributions

BD, KP, Conception and design, Acquisition of data, Analysis and interpretation of data, Drafting or revising the article; SH, CLF, Conception and design, Analysis and interpretation of data, Drafting or revising the article; TR-H, MKC, AF, RF, RRJ-C, JH, JT, SP, MWW, CMW, MDS, MD-R, Acquisition of data, Analysis and interpretation of data, Drafting or revising the article; JWo, JWi, JVO, CWM, JP, JL-T, PD, JFH, JT-O, EC, Analysis and interpretation of data, Drafting or revising the article, Contributed unpublished essential data or reagents; BMK, TH, RDEM, AK, ME, LKH, MC, RK, Acquisition of

data, Drafting or revising the article, Contributed unpublished essential data or reagents; JHH, MJC, Conception and design, Analysis and interpretation of data, Drafting or revising the article, Contributed unpublished essential data or reagents

## Author ORCIDs

Beatrice Demarchi, http://orcid.org/0000-0002-8398-4409
Michaela Ecker, http://orcid.org/0000-0001-9581-1882

# Additional files

## Supplementary files

• Supplementary file 1. Survival of ostrich eggshell proteins in time. The proteins identified in each ostrich eggshell sample are reported, together with the number of identified peptides and the percentage coverage, Val D/L value and hydropathicity.

• Supplementary file 2. Product ion spectra. Raw spectra (manually annotated on the basis of PEAKS assignments) of all the identified sequences identified in panel 1. 2–9: Copenhagen dataset; 10–26: York/Oxford dataset.

• Supplementary file 3. Diagenesis-induced modifications. Modifications detected in SCA-1 and SCA-2 sequences in all OES samples analysed.

• Supplementary file 4. Full proteomics dataset. This Excel file reports all the peptide and protein data for ostrich eggshell samples and the blanks.

## Major datasets

The following datasets were generated:

| Author(s) | Year | Dataset title | Dataset URL | Database, license, and accessibility information |
|---|---|---|---|---|
| Demarchi B | 2016 | Mineral-bound ostrich eggshell peptides survive for up to 3.8 Ma in equatorial Africa | http://www.ebi.ac.uk/pride/archive/projects/PXD003786 | Publicly available at EBI PRIDE (accession no. PXD003786) |
| Haile J | 2016 | BioProject ID PRJNA314978 | http://www.ncbi.nlm.nih.gov/biosample/SAMN04546447/ | Publicly available at NCBI BioSample (accession no: SAMN04546447) |
| Harding J, Hall S, Freeman C | 2016 | Simulations on calcite -protein interactions | http://dx.doi.org/10.15131/shef.data.3491387 | Publicly available at The University of Sheffield figshare (https://sheffield.figshare.com/) |

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

## Appendix 1

## Thermal age calculations

In this study we compared the thermal age of some of the oldest samples from which DNA and collagen sequence had been reported with the eggshell proteins from Africa (*Figure 1*). To estimate temperature fluctuation we estimated MATs for each site: each MAT was estimated from NOAA NCDC GCPS monthly weather station (*Eischeid et al., 1995*; *Karl et al., 1990*) and borehole data (*Huang et al., 2000*; *National Climatic Data Center (NCDC), 2012*) (*Appendix 1—Table 1*). Temperature was altitude-corrected using a lapse rate of 6.4°C / km$^{-1}$; where altitudes were not given they were estimated using the Google Maps API Elevation Service. For simplicity and due to the antiquity of the samples, no corrections were made for seasonal variation in temperature. Temperatures were then projected back using estimates of temperature fluctuation from the time of deposition to the present day from the long-term modelled temperature record of *Hansen et al. (2013)*. For the three youngest dates at Elands Bay Cave, the thermal ages were refined using the SST record from core MD02-2594 - Dyez14 (*Dyez et al., 2014*).

**Appendix 1—table 1.** Historical temperature data. Weather station temperature data derived from the NOAA Baseline Climatological Dataset - *Eischeid et al. (1995)*: The quality control of long-term climatological data using objective data analysis. Preprints of AMS Ninth Conference on Applied Climatology, Dallas, TX., January 15–20, 1995.

| WMO ID | Wmo station name | latitude | longitude | Alt. (m) | MAT °C |
|---|---|---|---|---|---|
| 7191700 | Eureka,N.W.T. | 79.98 | −85.93 | 10 | −19.6 |
| 100800 | Svalbard Lufthavn | 78.25 | 15.47 | 27 | −6.3 |
| 420201 | Dundas Radio Greenland | 76.6 | −68.8 | 20 | −10.5 |
| 420200 | Thule A.B. | 76.52 | −68.5 | 77 | −12.1 |
| 7195702 | Fort Mcpherson | 67.4 | −134.9 | 30 | −9.3 |
| 7196501 | Fort Selkirk | 62.8 | −137.4 | 454 | −3.9 |
| 358600 | Honington | 52.33 | 0.77 | 54 | 9.6 |
| 3605801 | Ust-Kan | 50.92 | 84.75 | 1037 | 0.6 |
| 807500 | Burgos/Villafria | 42.37 | 3.63 | 894 | 10.2 |
| 6832800 | Tsabong | −26.05 | 22.45 | 960 | 20.14 |
| 6871200 | Cape Columbine | −32.83 | 17.85 | 60 | 15.6 |
| 892800 | Mossel Bay (Cape St.) | −34.18 | 22.15 | 59 | 17.7 |
| **Borehole data: data from NOAA Paleoclimatology Borehole Datasets http://www.ncdc.noaa.gov/paleo/borehole** | | | | | |
| CA-289-2 | | 60.99 | −134 | 1524 | 0 |
| UK-STOWLANGTOFT | | 52.28 | 0.85 | 47 | 9.1 |
| ES-ROMANERA | | 37.69 | −7.33 | 166 | 20.2 |
| ES-AC-1BILLITON | | 37.6 | −6.83 | 110 | 18.7 |
| ES-PB1ADAROVALVERDE | | 37.56 | −6.78 | 237 | 18.6 |
| TZ-LONGIDO | | −2.61 | 36.47 | 1316 | 24.3 |
| TZ-BASOTU | | −4.38 | 35.17 | 1736 | 23.9 |
| TZ-KIZAGA | | −4.42 | 34.37 | 1472 | 23.2 |
| TZ-SIUYU | | −4.9 | 34.88 | 1678 | 23.1 |
| ZA-SB1 | | −27.28 | 25.5 | 1357 | 19.2 |
| ZA-AP11 | | −28.3 | 21.05 | 928 | 23.7 |
| ZA-PC227 | | −29.33 | 21.78 | 1052 | 21.5 |

**Appendix 1—table 2.** Long term climate records: cores used in this study to assess the extent of depression at LGM (used to scale).

| Record | latitude | longitude | Method | SST (°C) during LGM | Ref |
|---|---|---|---|---|---|
| ODP982 | 58 | −16 | UK37 | 13 | (*Lawrence et al., 2009*) |
| ODP882 | 50.35 | −167.58 | UK37 | 9.1 | (*Martínez-Garcia et al., 2010*) |
| MD02-2594 | −33.30 | 17.30 | Mg/Ca | ~16 | doi.pangaea.de/10.1594/PAN-GAEA.810663 |
| MD962077 | −33.17 | 31.25 | Mg/Ca | ~16 | doi.pangaea.de/10.1594/PAN-GAEA.810716 |
| ODP722A | 16.62 | 59.80 | UK37 | 20.2 | (*Herbert et al., 2010*) |
| ODP662 | −1.39 | −11.74 | UK37 | 18.7 | (*Herbert et al., 2010*) |
| ODP846 | −3.10 | −90.82 | Mg/Ca | 18.6 | (*Herbert et al., 2010*) |
| IODP1146 | 19.46 | 116.27 | UK37 | 24.3 | (*Herbert et al., 2010*) |

Because temperature changes likely increased with latitude and proximity to ice sheets the record of *Hansen et al. (2013)* was scaled to estimates of maximum decline in temperature for the site or the region at the last glacial maximum (LGM) (*Appendix 1—Table 4*). We estimate a fall of approximately 2.5°C at LGM in East Africa, consistent with observed reductions in Tex86 estimates of lake surface temperature reduction in Lake Tanganyika (*Tierney et al., 2010*) and Lake Malawi (*Konecky et al., 2011*). This decline of 2.5°C was used as the Δ°C LGM for Laetoli. Note that the offset between TEX86 and in situ temperature estimates in L. Tanganyika (*Kraemer et al., 2015*) does not alter estimates of Δ°C LGM. Late Pliocene global mean annual air temperature was estimated to have been 2–3°C warmer than today, but the increase was accentuated at higher latitudes. At Ellesmere Island (Beaver Pond) (*Ballantyne et al., 2010*) multiple proxies suggest Pliocene warming of +19 ± 1.9°C, significantly higher than mid-Pliocene simulations of the high Arctic (*Hill, 2015*).

**Appendix 1—table 3.** Kinetic parameters used for estimating thermal age.

| Target | Chemical reaction | Activation energy (kJ mol$^{-1}$) | Reference |
|---|---|---|---|
| Eggshell proteins | Valine racemization | 117 | (*Crisp et al., 2013*) |
| DNA | Depurination | 123 | (*Lindahl and Nyberg, 1972*) |
| Collagen | Gelatinization | 173 | (*Holmes et al., 2005*) |

Finally the thermal age (Time$_{yr@10°C}$) for each sample was calculated by examining the offset from 10°C at each time interval in the long-term temperature record and calculating the relative rate difference over this time interval using published kinetic parameters for the target molecule (*Appendix 1—Table 3*). For every year the site is warmer than 10°C, it accumulates thermal years faster than chronological years, while thermal years (yr@10°C) accrue more slowly for every year when the site is cooler than 10°C (e.g. *Figure 1*). To estimate the decomposition of ostrich eggshell proteins we used a conservative (low) activation energy, valine racemization (*Crisp et al., 2013*); the same diagenetic parameter used as the independent variable in e.g. *Figure 2*.

Thermal age highlights the differences in protein preservation between Ellesmere Island (MAT −19.7°C) and Laetoli (MAT 18.7°C), both sites which yield protein sequence, and both of which are estimated to be > 3.5 Ma. When corrected for thermal age, Laetoli is three orders of magnitude older than Ellesmere Island (*Appendix 1—Table 4*). This is actually a smaller difference than might be anticipated from the current difference between

**eLIFE** Research article

Biochemistry | Genomics and Evolutionary Biology

**Appendix 1—table 4.** Thermal age calculations.

| Site | Lat | Long | Alti (m) | MAT (°C) | Scale | ΔT LGM (°C) | Ref | Age (ka) | Rate | Ea (kJ) | ln A | Mean thermal age (Ma@10°C) | Min thermal age (Ma@10°C) | Max thermal age (Ma@10°C) |
|---|---|---|---|---|---|---|---|---|---|---|---|---|---|---|
| Ellesmere Island | 78.55 | −82.5 | 292 | −19.7 | 5.0 | −21 | (Ballantyne et al., 2010) | 3600 | Collagen | 173 | 61 | 0.003 | 0.001 | 0.007 |
| Ellesmere Island | 78.55 | −82.5 | 292 | −19.7 | 5.0 | −21 | (Ballantyne et al., 2010) | 3800 | Collagen | 173 | 61 | 0.004 | 0.001 | 0.011 |
| Poolepynten | 79 | 11.43 | 4 | −6.3 | 4.0 | −16 | (Hubberten, 2004) | 130 | DNA | 127 | 41.2 | 0.002 | 0.002 | 0.002 |
| Thistle Creek | 63.11 | −139.54 | 483 | −4.1 | 1.0 | −4 | (Elias, 2001) | 560 | DNA | 127 | 41.2 | 0.02 | 0.02 | 0.02 |
| Thistle Creek | 63.11 | −139.54 | 483 | −4.1 | 1.0 | −4 | (Elias, 2001) | 700 | DNA | 127 | 41.2 | 0.03 | 0.03 | 0.03 |
| Happisburgh | 52.82 | 1.53 | 2 | 9.1 | 2.0 | −9 | (Harrison et al., 2015) | 700 | Collagen | 173 | 61 | 0.22 | 0.19 | 0.25 |
| Sima de los Huesos | 42.35 | −3.52 | 1004 | 9.5 | 1.0 | −4 | (Moreno et al., 2012) | 400 | DNA | 127 | 41.2 | 0.25 | 0.22 | 0.28 |
| Elands Bay Cave | −32.32 | 18.32 | 10 | 15.7 | 1.0 | −4 | (Martínez-Méndez, 2010) | 16 | Valine | 118 | 38 | 0.04 | 0.03 | 0.05 |
| Pinnacle Point | −34.2 | 22.09 | 20 | 18 | 1.0 | −4 | (Bar-Matthews et al., 2010) | 150 | Valine | 118 | 38 | 0.42 | 0.38 | 0.47 |
| Wonderwerk | 27.8 | 23.55 | 1645 | 18.6 | 0.6 | −2.5 | (Ecker et al., 2015) | 1000 | Valine | 118 | 38 | 3.6 | 3.1 | 4.1 |
| Olduvai | −2.99 | 35.35 | 576 | 26 | 0.6 | −2.5 | (Tierney et al., 2010) (Berke et al., 2012) | 1338 | Valine | 118 | 38 | 16.3 | 14.4 | 18.5 |
| Laetoli | −3.22 | 35.19 | 1700 | 18.7 | 0.6 | −2.5 | (Tierney et al., 2010) (Berke et al., 2012) | 3800 | Valine | 118 | 38 | 16.0 | 13.6 | 18.7 |

the sites (MAT Δ 39°C), as despite high latitude Pleistocene glacial stage cooling, Ellesmere Island was much warmer during the Pliocene (*Ballantyne et al., 2010*).

## Appendix 2

# Ostrich eggshell bleached proteome

### Microstructure of OES layers

We characterised the microstructure and the proteins extracted from modern *S. camelus* eggshell. The whole eggshell had a dimension of 25 cm in length with a diameter of 15 cm and an average thickness of 2 mm. We also examined one specimen of fossil eggshell from Laetoli: LOT 13898, a fragment.

The modern OES was first sectioned in three pieces using a diamond-blade saw. Cross-sections of approx. $10 \times 4 \times 2/3$ mm in size were cut off with a wheel for SEM observations. Each cross section was manually polished down to approximately 50 µm thickness using a diamond lapping pad (30 to 0.1 µm) and a tripod holder, allowing a transversal view of each individual layer. The surface was slightly etched using 2% acetic acid to provide topography to the surface. The cross and transversal sections were mounted in a SEM aluminium stub covered by a sticky carbon pad. All samples were coated with a 10 nm Pt/Pd layer. Further, a subsample from each modern OES layer was prepared by polishing off the rest of the layer and then each individual layer was powdered and bleached for amino acid analyses. For the modern OES a FEI Sirion FEG FESEM (field emission, high resolution Scanning Electron Microscope) and for the Laetoli OES a JEOL JSM-7800 were used, both operated at 5 kV for imaging, and 5 and 10 mm working distance respectively.

The modern OES presented a well-defined structure consisting of three structural calcitic layers, as observed by others (*Heredia et al., 2005*; *Figure 2—figure supplement 1*). The external layer (in contact with the environment) is the thin crystalline layer with little organic content (*Heredia et al., 2005*). The middle layer is the palisade, and the innermost layer is the mammillary/cone layer (*Feng et al., 2001*; *Patnaik, 2009*). The palisade and cone layers contain organic components, both protein (*Hincke et al., 1995*) and polysaccharides (*Baker and Balch, 1962*), which help hold the calcite crystals together (*Heredia et al., 2005*). The SEM microphotograph of the cross section of the eggshell showed that the thickness of the crystal layer is about 40 µm, the palisade layer is the thicker with a thickness of 1200 µm and cone layer has a thickness of 750 µm. A deeper understanding of the microstructure of each individual layer is gained by observations in the direction perpendicular to the cross section. The crystalline layer is very compact and dense with porosity distributed along its surface. The palisade layer consists of an agglomeration of flakes more or less parallel to the surface of the eggshell; the cone layer consists of a pile of elongated crystals perpendicular to the surface of the eggshell; the organic membrane consists of a net of different fibres which lie parallel to the eggshell surface. Sample LOT 13898 from Laetoli (*Figure 2—figure supplement 1A*, right), attributed to *S. kakesiensis*, is 33% thicker than the modern *S. camelus* eggshell. It has been reported before (*Harrison and Msuya, 2005*) that samples from the Upper Laetolil beds present a thickness ranging from 2.5 to 4.4 mm. In this sample, the organic layer is not present, probably due to degradation after deposition.

### AAR analyses on OES layers (modern)

Total hydrolysable amino acids (THAA) were analysed on one bleached and one unbleached modern eggshell subsample per microstructural layer (*Appendix 2—table 1*). While the absolute concentrations were higher in unbleached eggshell for all layers (*Figure 2—figure supplement 1B*), the overall amino acid composition is comparable between (a) bleached and unbleached powders and (b) the different microstructural layers (*Figure 2—figure supplement 1C*). This points towards a similar composition of the proteome enclosed in the intracrystalline fraction of the three main microstructural layers of OES.

## Bleached OES proteomes characterized by LC-MS/MS

66 LC-MS/MS analyses were conducted as part of this study (York/Oxford dataset): 33 bleached ostrich eggshell subsamples and 33 blanks. 71161 product ion spectra were matched to known peptide sequences (excluding common contaminant proteins).

## Ostrich eggshell proteome

The modern bleached OES yielded 273 unique protein descriptions.(*Appendix 2—Table 2*) reports the most abundant protein groups, i.e. the ones identified on the basis of 20 or more unique peptides. A Gene Ontology (GO) analysis showed that most proteins' function, cellular component and biological process are presently unknown, while 26 proteins are associated with binding and 14 with catalytic activities.

Here we summarize the roles and main characteristics of the most abundant protein groups found in the bleached eggshell proteome.

- **Struthiocalcin-1 and struthiocalcin-2** are mineralization-specific C-type lectins originally isolated and sequenced from OES matrices (*Mann and Siedler, 2004*); their structure and role in eggshell biomineralization have been recently investigated (*Sánchez-Puig, 2012*; *Ruiz-Arellano and Moreno, 2014*; *Ruiz-Arellano et al., 2015*).

- **Aggrecan** proteins are important constituents of the cartilage and this large aggregating chondroitin sulfate proteoglycan binds to hyaluronic acid via an N-terminal globular region and may regulate the matrix assembly of the cartilage (*Kiani et al., 2002*; *Watanabe et al., 1998*).

- **Vitelline membrane outer layer protein 1** (**VMO1**): the vitelline membrane outer layer in hens is formed in the upper oviduct after ovulation. This membrane is a fibrous layer constructed from ovomucin (with soluble proteins bound) and proteins vitelline membrane outer layer I and II (*Shimizu et al., 1994*).

- **Serum albumin** (**ALB**) is the main protein of plasma, has a good binding capacity for water, Ca2+, Na+, K+, fatty acids, hormones, bilirubin and drugs. Its main function is the regulation of the colloidal osmotic pressure of blood (UniProtKB - P19121 (ALBU_CHICK)). It has been identified as abundant in the eggshell of chicken, turkey and quail (*Mann and Mann, 2015*).

- The **tenascin** (**TNC**) family of glycoproteins is expressed in the embryo, particularly during neural development, skeletogenesis, and vasculogenesis, and contains a serine-proline rich domain (*Jones and Jones, 2000*). They appear to accelerate collagen fibril formation and may play a role in supporting the growth of epithelial tumours (*Egging et al., 2007*).

- **Carbonic anhydrase** is a family of enzymes with diverse amino acid sequences and structures that catalyse the slow conversion between carbon dioxide and bicarbonate (*Liljas and Laurberg, 2000*). The presence of this enzyme in the hen's reproductive tract was linked very early to its role in making carbonate ions available for eggshell formation (*Robinson and King, 1963*; *Gutowska and Mitchell, 1945*).

- **Mucins** (**LOC 100859916**)are gel-forming proteins that have been linked to biomineralization in mollusc shells (*Marin et al., 2000*) and have been identified in eggshell membranes, most recently in a proteomics study which identified a 1–10 fold increase of mucin 5AC in fertilized egg membranes (*Cordeiro and Hincke, 2016*).

- **Apolipoprotein D** (**APOD**)is expressed in subsets of central nervous system neurons and glia during late chicken embryogenesis (*Ganfornina et al., 2005*). APOD can bind cholesterol, progesterone, pregnenolone, bilirubin and arachidonic acid in plasma and may be involved in repairing the nervous system (*Rassart et al., 2000*).

- The antibody **immunoglobulin** from egg exists as IgG (or IgY) in yolk and as two forms in egg white (IgA, IgM) (*Abdou et al., 2013*). The **polymeric immunoglobulin receptor** is involved in the secretion of antibodies and was one of the proteins identified in the acid-

**Appendix 2—table 1.** THAA concentrations (pmol/mg) detected in OES microstructural layers; bleached and unbleached powders were analysed. Analytical errors measured on replicate analyses are < 5% (Asx = 3.80%; Glx = 3.81%; Gly = 4.89%; Ala = 3.67%; Val = 3.98%; Ile = 3.63%).

| | [Asx] | [Glx] | [Ser] | [L-Thr] | [L-His] | [Gly] | [L-Arg] | [Ala] | [Tyr] | [Val] | [Phe] | [Leu] | [Ile] |
|---|---|---|---|---|---|---|---|---|---|---|---|---|---|
| Cone-bleached | 4925 | 5409 | 3992 | 2589 | 1520 | 6923 | 2655 | 4314 | 1579 | 2815 | 1983 | 4264 | 2280 |
| Cone-unbleached | 14452 | 17835 | 13455 | 9140 | 4894 | 19136 | 8108 | 10234 | 4342 | 9734 | 4334 | 10500 | 7676 |
| Palisade-bleached | 5990 | 6434 | 4479 | 2347 | 2079 | 6641 | 3427 | 5183 | 1525 | 2768 | 2659 | 5032 | 3052 |
| Palisade-unbleached | 10018 | 11545 | 7967 | 4507 | 3667 | 12691 | 6247 | 9085 | 3052 | 4843 | 4505 | 8446 | 5102 |
| Whole-bleached | 6323 | 6533 | 4662 | 2445 | 2032 | 6976 | 3322 | 5314 | 1263 | 2943 | 2732 | 5289 | 3177 |
| Whole-unbleached | 12412 | 13774 | 9927 | 5420 | 4216 | 14951 | 7222 | 10670 | 4364 | 6003 | 5342 | 10152 | 6342 |
| Crystalline-bleached | 4518 | 4181 | 3085 | 1301 | 1399 | 4097 | 1981 | 3528 | 49 | 1704 | 1838 | 3299 | 2225 |
| Crystalline-unbleached | 24422 | 24212 | 17990 | 8100 | 6748 | 23365 | 12371 | 20055 | 6445 | 9912 | 10755 | 19079 | 12535 |

soluble organic matrix of the chicken calcified eggshell layer additionally to the immunoglobulins (**Mann et al., 2006**).

• **Mesothelin** (**MSLNL**) is expressed in eggshell membranes of fertilized chicken eggs (**Cordeiro and Hincke, 2016**). This cell-surface differentiation antigen is normally expressed at low levels in humans and is restricted to tissues such as the mesothelial cells lining some body cavities and epithelial cells of kidney, tonsil, trachea, and fallopian tube; it is overexpressed in and represents a marker for a range of tumours, including human and hen ovarian cancer (**Yu et al., 2011**).

• **Golgi apparatus protein 1** or cysteine-rich fibroblast growth factor receptor was isolated in chick embryos (**Burrus and Olwin, 1989**); it is located in the Golgi and may be involved in intracellular fibroblast growth factor trafficking and the regulation of cellular responses to these (**Zuber et al., 1997**).

• **Stanniocalcin-1** is a member of the stanniocalcins family, present in all vertebrates and linked to calcium homeostasis but also to embryogenesis and tumorigenesis (**Trindade et al., 2009**).

• **Ovomucoid** (**IOVO**) is one of the major egg white proteins; it is a glycoprotein protease inhibitor well-characterized in chicken, turkey and quail (**Mann and Mann, 2015**).

• **Serotriflin** belongs CRISP family protein with binding affinity for small serum protein-2 in snake serum (**Aoki et al., 2008**). BLAST analysis shows that serotriflin-like proteins have been identified in the genomes of 39 Archosauria. In ostrich the cysteine-rich secretory protein 2 (gene: N308_13534) displays 54% identity with snake serotriflin (P0CB15 CRIS_PROFL).

• **Delta and Notch-like epidermal growth factor-related receptor** (**DNER**)is a calcium-binding clathrin-binding protein linked to central nervous system development (GO term).

• **BPI** (bactericidal permeability-increasing) **fold-containing family B member 4** is a lipid-binding protein identified in fertilized eggshell membrane of chicken (**Cordeiro and Hincke, 2016**). It shares sequence homology with ovocalyxin-36 (OCX-36), which has an immune role (**Cordeiro et al., 2013**). In *S. camelus* this protein (gene: N308_11956) shares only 37% identity with chicken OCX-36 (BLASTp), although this is likely due to missing regions of OCX-36. BLAST analysis also identified the **uncharacterized protein LOC104140623** as a homologue to BPI-fold-containing family B member 4.

• **Pigment epithelium-derived factor** (PEDF) or serpin F1 (SERPINF1) is a non-inhibitory serpin and neurite-promoting factor. Human SERPINF1 and ostrich PEDF share 63% identity (BLAST analysis).

• **Prosaposin** (P07602|SAP_HUMAN) shares 56% identity with *Struthio* proactivator polypeptide (saposin A-D precursor); prosaposin is also found in chicken eggshell cuticle (**Rose-Martel et al., 2012**) and is abundant in the calcified eggshell of quail, turkey and chicken (**Mann and Mann, 2015**). The saposins are involved in lysosomal sphingolypid metabolism and bind glycolipids (**Mann and Mann, 2015**).

• **Pantetheinase-like isoform X1** (gene N308_06452 in *Struthio camelus australis*, 67% identity with human pantetheinase) is an amidohydrolase involved in the dissimilative pathway of CoA (**Maras et al., 1999**).

• **Beta-microglobulin** (**B2M**) is involved in folding of the eggshell matrix proteins (**Nys et al., 2011**) and was identified in the calcified chicken eggshell (**Mann et al., 2006**) and in the membranes (**Cordeiro and Hincke, 2016**).

• **Cygnin** is a small basic protein found in 62 bird genomes and analogue to Duck Basic Protein Small 2 isolated from duck egg white and shown to perform multiple biological functions related to reducing the risk of diseases usingin vitro experiments (**Naknukool et al., 2011**).

• **Signal peptide CUB and EGF-like domain-containing protein 1 partial** (**SCUBE1**) binds calcium and may function as an adhesive molecule; it is involved in post-embryonic

development and the *Struthio* analogue (**gene: N308_04062**) shares 83% of its sequence with human SCUBE1.

- **Ovalbumin** (**OVAL**) is abundant both in egg white, in eggshell membranes and the calcified layers of a range of birds (*Mann et al., 2006*; *Mann and Mann, 2015*; *Cordeiro and Hincke, 2016*). Ovalbumin was found to stabilize the liquid precursor phase of calcium carbonate (*Wolf et al., 2011*).

- **Ovostatin-like** (ovomacroglobulin) in *Struthio* shares 80% of its sequence with chicken ovostatin (OVST). Ovostatin is a proteinase inhibitor which reduces enzyme activity in the presence of high-molecular weight substrates.

**Appendix 2—table 2.** 30 major protein groups identified in bleached modern OES (>20 unique peptides only).

| Protein group | Accession | -lgP | Coverage (%) | #Unique | Description |
|---|---|---|---|---|---|
| 1 | gi\|46396750 | 344 | 100 | 346 | RecName: Full = Struthiocalcin-1; Short = SCA-1 |
| 2 | gi\|46396751 | 293 | 98 | 222 | RecName: Full = Struthiocalcin-2; Short = SCA-2 |
| 7 | gi\|697501075, gi\|678217626 | 281 | 39 | 208 | Aggrecan core protein [*Struthio camelus australis*] |
| 8 | gi\|697508924 | 268 | 88 | 126 | Vitelline membrane outer layer protein 1-like [*Struthio camelus australis*] |
| 10 | gi\|697509029 | 268 | 80 | 113 | Serum albumin-like [*Struthio camelus australis*] |
| 13 | gi\|697455783 | 246 | 40 | 89 | Tenascin isoform X3 [*Struthio camelus australis*] |
| 11 | gi\|697477202 | 234 | 75 | 84 | Carbonic anhydrase 4 [*Struthio camelus australis*] |
| 14 | gi\|697481828 | 237 | 15 | 82 | Mucin-5AC [*Struthio camelus australis*] |
| 9 | gi\|697523391, gi\|678221588 | 228 | 69 | 77 | Apolipoprotein D [*Struthio camelus australis*] |
| 12 | gi\|375162648 | 231 | 75 | 76 | Immunoglobulin A heavy chain constant region secretory form partial [*Struthio camelus*] |
| 15 | gi\|697470179, gi\|697470177 | 213 | 51 | 63 | Mesothelin isoform X1 [*Struthio camelus australis*] |
| 22 | gi\|697433909, gi\|678206587 | 195 | 40 | 45 | Golgi apparatus protein 1 [*Struthio camelus australis*] |
| 17 | gi\|375162644 | 192 | 63 | 42 | immunoglobulin M heavy chain constant region secretory form partial [*Struthio camelus*] |
| 19 | gi\|678209093, gi\|697441180 | 197 | 66 | 41 | Stanniocalcin-1 partial [*Struthio camelus australis*] |
| 18 | gi\|697514088, gi\|678219803 | 197 | 86 | 39 | Ovomucoid [*Struthio camelus australis*] |
| 20 | gi\|697488611 | 191 | 89 | 39 | Serotriflin-like [*Struthio camelus australis*] |
| 21 | gi\|678214778 | 192 | 42 | 38 | Delta and Notch-like epidermal growth factor-related receptor partial [*Struthio camelus australis*] |
| 23 | gi\|697475278, gi\|697475274, gi\|697475280 | 192 | 41 | 37 | Polymeric immunoglobulin receptor [*Struthio camelus australis*] |
| 26 | gi\|697430975 | 181 | 33 | 33 | BPI fold-containing family B member 4-like [*Struthio camelus australis*] |

*Appendix 2—table 2 continued on next page*

*Appendix 2—table 2 continued*

| Protein group | Accession | -lgP | Coverage (%) | #Unique | Description |
|---|---|---|---|---|---|
| 25 | gi\|697430936 | 161 | 42 | 32 | Uncharacterized protein LOC104140623 [*Struthio camelus australis*] |
| 31 | gi\|697485873, gi\|678214846 | 172 | 68 | 28 | Pigment epithelium-derived factor [*Struthio camelus australis*] |
| 28 | gi\|697430934, gi\|678205748 | 153 | 34 | 26 | BPI fold-containing family B member 4-like [*Struthio camelus australis*] |
| 29 | gi\|697432918 | 172 | 41 | 26 | Prosaposin [*Struthio camelus australis*] |
| 16 | gi\|375162654 | 178 | 96 | 24 | immunoglobulin lambda constant region partial [*Struthio camelus*] |
| 30 | gi\|697522911, gi\|697522913, gi\|697522916 | 172 | 50 | 22 | Pantetheinase-like isoform X1 [*Struthio camelus australis*] |
| 24 | gi\|697446419 | 165 | 71 | 22 | Beta-2-microglobulin [*Struthio camelus australis*] |
| 27 | gi\|678210026,gi\|678210025 | 154 | 63 | 22 | Cygnin [*Struthio camelus australis*] |
| 34 | gi\|678216365 | 173 | 31 | 21 | Signal peptide CUB and EGF-like domain-containing protein 1 partial [*Struthio camelus australis*] |
| 32 | gi\|697492053 | 143 | 42 | 21 | Ovalbumin {ECO:0000303\|PubMed:21058653} [*Struthio camelus australis*] |
| 37 | gi\|697505689 | 147 | 13 | 20 | Ovostatin-like [*Struthio camelus australis*] |

## Common contaminants in bleached OES

The common contaminant proteins identified in bleached OES samples and the procedural blanks are reported in *Appendix 2—Table 3*. The enzymes trypsin and elastase are detected in all samples except in the Laetoli subsamples: one of these was prepared without digestion step (13901N), while all the other subsamples were digested with trypsin or elastase but analysed separately by LC-MS/MS. Keratins were also detected in all samples, despite the thorough cleaning procedures adopted (including a long bleaching step). This highlights the need for careful evaluation of ancient protein data from fossil samples.

**Appendix 2—table 3.** List of contaminant proteins detected in the 66 analyses, with total number of spectra identified per protein.

| Row labels | Count of #Spec |
|---|---|
| sp\|TRYP_PIG\| | 771 |
| sp\|K2C1_HUMAN\| | 493 |
| Chymotrypsin-like elastase family member 1 OS = Sus scrofa GN = CELA1 PE = 1 SV = 1 | 422 |
| sp\|K1C9_HUMAN\| | 301 |
| sp\|K1C10_HUMAN\| | 216 |
| sp\|K22E_HUMAN\| | 213 |
| sp\|TRFE_HUMAN\| | 98 |
| sp\|ALBU_HUMAN\| | 94 |
| PREDICTED: keratin type II cytoskeletal cochleal isoform X1 [*Struthio camelus australis*] | 80 |
| PREDICTED: keratin type II cytoskeletal cochleal isoform X2 [*Struthio camelus australis*] | 32 |
| PREDICTED: keratin type II cytoskeletal 5-like [*Struthio camelus australis*] | 25 |
| sp\|RS27A_HUMAN\| | 23 |
| Keratin type II cytoskeletal 75 partial [*Struthio camelus australis*] | 20 |

*Appendix 2—table 3 continued on next page*

*Appendix 2—table 3 continued*

| Row labels | Count of #Spec |
| --- | --- |
| sp|GFP_AEQVI| | 19 |
| sp|TRY1_BOVIN| | 19 |
| Keratin type II cytoskeletal cochleal partial [*Struthio camelus australis*] | 16 |
| PREDICTED: keratin type II cytoskeletal cochleal isoform X3 [*Struthio camelus australis*] | 16 |
| sp|ANT3_HUMAN| | 8 |
| sp|HBB_HUMAN| | 8 |
| Keratin type II cytoskeletal 75 [*Struthio camelus australis*] | 5 |
| sp|TRFL_HUMAN| | 5 |
| Keratin type I cytoskeletal 14 partial [*Struthio camelus australis*] | 3 |
| PREDICTED: keratin type II cytoskeletal 4-like [*Struthio camelus australis*] | 3 |
| sp|HBA_HUMAN| | 3 |
| sp|K1C15_SHEEP| | 2 |
| sp|GSTP1_HUMAN| | 1 |

## Appendix 3

## Computational

Conventional molecular dynamics (MD) was used to explore possible protein–calcite binding geometries. In order to determine the most likely structure of SCA-1 and SCA-2 in solution, initial structural estimates were obtained from crystal structures. The model structure for SCA-2 was obtained using a homology model (*Biasini et al., 2014*) based on the published crystal structure (*Ruiz-Arellano et al., 2015*). Calcite structures were optimized using the Raiteri forcefield (*Raiteri et al., 2010*), selected due to its parameterization against the free energy of dissolution of calcium carbonate, which provides reliable descriptions of the structuring and interactions between water and the surface. The resultant structures possessed the known density of calcite, 2.71 g/mL, and their exposed surfaces maintained the (10.4) structure. Since the calcite blocks were of sufficient thickness, it was not necessary to freeze the atoms in the middle of the calcite block. Charges for SCA-1 and SCA-2 were calculated using Mulliken charges obtained from the AMBER Antechamber program (*Wang et al., 2004*). The overall charges for SCA-1 and SCA-2 were determined to be −11 and −10 respectively using the Avogadro code (*Hanwell et al., 2012*) at pH 8.0 to 10, the point of zero charge for calcite. Cross terms for the interaction of the atoms within the protein and peptide molecules were calculated using Lorentz-Berthelot mixing rules (*Lorentz, 1881*; *Berthelot, 1898*); cross terms for interactions between the protein and peptide molecules and the calcium carbonate surface were taken from the Raiteri potential (*Raiteri et al., 2010*) or otherwise fitted via scaling methods based on the charge differences of the ions as described previously in the literature (*Freeman et al., 2007*; *Schröder et al., 1992*). A mean cut-off of 12 Å was used for van der Waals interactions. All calculations were performed using the DL POLY Classic program (*Todorov et al., 2006*).

The surface unit cell for the (10.4) surface was defined by two perpendicular vectors of length 83.544 Å, which gives a total surface area of 6979 Å$^2$, sufficient to accommodate the proteins or the peptide sequences. A block of 10 layers of calcite was used (giving a thickness of 32 Å). The water layer was a depth of 82 Å (about 30,000 water molecules), which is sufficient for an adequate coverage of the biomolecules.

Since performing full adsorption studies of these proteins in every possible binding configuration was not computationally tractable, it was necessary to perform an initial study of the relative binding energies of the proteins placed at the surface in various configurations. This was accomplished by performing Euler transformations on the proteins, rotating them in 5 degree increments over the angles alpha, beta, and gamma from 0 to 180 degrees. The reoriented protein was then placed at the calcite surface with 30,000 TIP3P water molecules, providing a water density of 0.99 g/mL, and subjected to an energy minimization routine in DL POLY Classic with the calcite block held rigid. Five calcium counterions were placed in each solution with one additional sodium counterion needed for SCA-1.

This enabled us to choose low energy configurations for more detailed investigation. Binding energy calculations were performed using the following set of simulations upon each: protein/amino acid sequence in aqueous solution, protein/amino acid sequence starting in an unbound position above the surface and adsorbing onto the surface and a simulation of the protein/amino acid sequence in aqueous solution bound to the surface. If compared to simulations of the calcite-water interface and pure water itself, it is possible to obtain information based solely upon the interaction of these proteins/amino acid sequences and the surfaces. The following equations were used, employing configurational energies in all cases:

$$E_{\text{(aqueous)}} = E_{\text{(molecule; aqueous)}} - E_{\text{(water)}} - E_{\text{(molecules)}} \tag{1}$$

$$E_{\text{(molecule; binding)}} = E_{\text{(molecule;hydrated surface)}} - E_{\text{(hydrated surface)}} - E_{\text{(molecule; aqueous)}} + E_{\text{(water)}} \tag{2}$$

In **Equations 1 and 2** the meaning of the symbols is as follows: $E_{\text{(aqueous)}}$ is the solvation energy of the protein/amino acid sequence in water; $E_{\text{(molecule; aqueous)}}$ is the total energy of the molecule in water; $E_{\text{(water)}}$ is the energy of a box of water containing the same number of water molecules as the calculation for $E_{\text{(molecule; aqueous)}}$ to ensure **Equation (2)** is balanced; and $E_{\text{(molecule)}}$ is the energy of the protein/amino acid sequence molecule in vacuum; $E_{\text{(molecule; binding)}}$ is the energy of binding of the molecule to the surface; $E_{\text{(molecule; hydrated surface)}}$ is the total energy of the molecule on the surface in the presence of water; $E_{\text{(hydrated surface)}}$ is the energy of the hydrated surface with the same number of calcium carbonate units as the previous calculation.

Additionally we have estimated the entropic contribution to the molecular binding arising from the displacement of water molecules from the surface to bulk water. From our previous work (**Freeman and Harding, 2014**) we expect this to be the dominant contribution to the entropy of binding for molecules of this kind. Our results show that the number of water molecules displaced is in the range 20.2–23.1 for the proteins and 6.6–8.3 for the amino acid sequences. This corresponds to an entropic contribution of 36.4–41.6 kJ mol$^{-1}$ for the proteins and 11.8–14.9 kJ mol$^{-1}$ for the amino acid sequences at 300 K, sufficiently small to justify the use of configurational energies in this work to estimate the strength of binding. It should also be noted that a smaller, negative contribution to the entropy is expected from the reduced freedom of motion of the protein/amino acid sequences at the calcite surfaces, which is considered to be constant between all binding profiles and expected to be smaller than that of many water molecules.

Simulations of these protein/amino acid sequences in water were performed by studying each of the systems for 10 ps in steps of 10 K from 10 K to 330 K. The lowest energy configurations of these proteins/amino acid sequences in aqueous solution were obtained from averages of 3 separate simulations over 3 ns of simulation time post equilibration at a temperature slightly above that used for other experiments, 330 K, which was selected as an efficient method to explore configurations at energies near the lowest energy configuration without having to resort to more expensive methods such as replica exchange or metadynamics. These structures were utilized for further calculations on surfaces, and for calculations in aqueous solution at 300 K. It was found that at this given temperature these protein/amino acid sequences would adopt a low energy conformation within approximately 500 ps and would not vary in energy by more than 5% of the total energy during the remainder of the simulation. 3 ns of simulation time at 300 K with a 1 fs timestep was sufficient to provide acceptable statistics for analysis. The simulations were performed at a constant volume (NVT ensemble) employing a Nose-Hoover thermostat with a relaxation time of 1 ps. For the systems containing surfaces, the lowest energy configuration of each protein or peptide sequence is placed such that the nearest atom to the surface is within 3 Å of the surface to ensure absorption to the surface within a reasonable timeframe (accounting for the slow kinetics of molecules diffusing through the adsorbed water layers) and again simulated for 3 ns. Molecules were placed at this distance due to the large barriers to crossing the organized water layers at the surface.

Water residence times were calculated and averaged within 1.1 Å regions surrounding atoms comprising each individual amino acid within the sequences identified as being most likely to be present at the surface during protein binding to calcite. Water residence times were calculated by fitting survival probability functions as defined below:

$$P_\alpha(t) = \sum_{j=1}^{N_w} \frac{1}{N-m+1} \sum_{n=1}^{m} p_{\alpha j}(t_0, t_0 + t', \delta t) \qquad (3)$$

where $t = m\delta t$ and $t' = n\delta t$. The binary function $P_{\alpha j}(t_0;\ t_0 + t';\ \delta t)$ takes the value of 1 when the water molecule remains within the shell defined by $\alpha$ during both times $t_0$ and $t_0 + t'$ and is otherwise equal to zero. $t$ is set to 0.1 ps; $N$ is the total number of configurations produced during the molecular dynamics simulation and $N_w$ is the total number of water molecules in the system. Average residence times can be obtained by fitting these functions to a simple exponential assuming a single relaxation time, $P_\alpha t = P_0 \exp(-t/\tau)$. The values were further averaged for each amino acid in the relevant amino acid sequences in order to get an estimate of how much each amino acid within the sequence affected water mobility within its sphere of influence.

## Loss of entropy of water molecules

We provide a conservative estimate of the loss of entropy due to surface stabilization in the main text. However, this might be even more significant than estimated, as suggested by further computational examination of the water on the mineral surface around the peptides, which shows that this value may underestimate the total entropy loss of the water molecules. The average residence time for the water molecules (120 ps) increases in the presence of the peptides (*Table 2*), particularly with the tightest binding peptide sequence (with the DDDD component) to $135 \pm 3$ ps. We also examined the self-diffusion coefficients of the water molecules in the different environments present within these simulations. In a simulation of pure water, the diffusion coefficient for TIP3P water is approximately $6 \times 10^{-9}$ m$^2$s$^{-1}$, a value which drops significantly for molecules bound near the surface to approximately $1 \times 10^{-9}$ m$^2$s$^{-1}$. For water molecules trapped between SCA-1 or its substituent peptides and the calcite surface, this diffusion coefficient is of the order of $1 \times 10^{-10}$ m$^2$s$^{-1}$.

## Appendix 4

> ## Degradation patterns
>
> ### A. AAR on archaeological OES
>
> Here we report the results of chiral amino acid analyses on all OES samples included in this study. All FAA and THAA D/L values increase with increasing age of the samples (estimated from the numerical dates obtained on the archaeological layers and reported in *Tables 3– 7*). The extent of hydrolysis (estimated as the percentage of free amino acids: FAA = [FAA]/ [THAA]*100) also increases with time, although the values are more variable than the DL ratios, mainly due to imprecisions in the measurement of the small masses and volumes involved. (*Appendix 4—table 1* and *2*) report the amino acid racemization (D/Ls and % FAA data) for all samples analysed in this study. Thermal age estimates are also reported for each sample.
>
> **Appendix 4—table 1.** D/L values of archaeological OES samples analysed in this study. b = bleached; H* = THAA obtained by 24-hr acid hydrolysis; F = FAA. Ile A/I values could not be calculated for Laetoli samples due to the presence of a compound co-eluting with D-alloisoleucine. Analytical errors measured on replicate analyses are <5% (D/Ls: Asx = 0.54%; Glx = 1.18%; Ala = 3.94%; Val = 2.09%; Ile = 3.77%. Concentrations: Asx = 3.80%; Glx = 3.81%; Gly = 4.89%; Ala = 3.67%; Val = 3.98%; Ile = 3.63%). (*H*) Sample 4605 yielded high D/L values because this sample had been exposed to high temperatures in the burial environment (burning [*Crisp, 2013*]). Note: thermal age calculations were performed on the basis of the Hansen model (*Hansen et al., 2013*); due to the absence of more continuous record for younger (last 2000 years) samples in the Hansen record, the Elands Bay Cave time points <1600 years refined using the SST record from core MD02-2594 - Dyez14 (Dyez et al. 2014).
>
> | LOT | NEaar | Asx D/L | Glx D/L | Ala D/L | Val D/L | Ile A/I | Thermal age (years) |
> |-----|-------|---------|---------|---------|---------|---------|---------------------|
> | 1868 | 6887bH* | 0.291 | 0.061 | 0.066 | 0.075 | 0.052 | 401−564 |
> | 1868 | 6887bF | 0.346 | 0.074 | 0.119 | 0.098 | 0.076 | |
> | 1872 | 6888bH* | 0.268 | 0.056 | 0.061 | 0.025 | 0.032 | 1313–1932 |
> | 1872 | 6888bF | 0.212 | 0.079 | 0.100 | 0.000 | 0.000 | |
> | 1866 | 6889bH* | 0.278 | 0.057 | 0.066 | 0.028 | 0.052 | 3962−5704 |
> | 1866 | 6889bF | 0.331 | 0.062 | 0.116 | 0.000 | 0.076 | |
> | 1849 | 6891bH* | 0.414 | 0.095 | 0.142 | 0.053 | 0.081 | 19,759–27,951 |
> | 1849 | 6891bF | 0.555 | 0.136 | 0.226 | 0.138 | 0.169 | |
> | 1850 | 6893bH* | 0.524 | 0.141 | 0.215 | 0.080 | 0.133 | 19,759–27,951 |
> | 1850 | 6893bF | 0.698 | 0.177 | 0.368 | 0.166 | 0.254 | |
> | 1823 | 6896bH* | 0.462 | 0.117 | 0.185 | 0.073 | 0.091 | 25,657–36,249 |
> | 1823 | 6896bF | 0.621 | 0.161 | 0.301 | 0.158 | 0.208 | |
> | 1819 | 6899bH* | 0.714 | 0.222 | 0.317 | 0.112 | 0.184 | 26,878–37,877 |
> | 1819 | 6899bF | 0.806 | 0.294 | 0.442 | 0.217 | 0.332 | |
> | 1840 | 6907bH* | 0.469 | 0.123 | 0.187 | 0.095 | 0.122 | 32,379–44,863 |
> | 1840 | 6907bF | 0.665 | 0.171 | 0.327 | 0.168 | 0.241 | |
> | 4605(*H*) | 7198bH* | 0.866 | 0.717 | 0.840 | 0.525 | 0.708 | 166,094–198,795 |
> | 4605(*H*) | 7198bF | 0.931 | 0.782 | 0.931 | 0.692 | 0.955 | |
> | 4613 | 7676bH* | 0.661 | 0.226 | 0.411 | 0.210 | 0.269 | 122,139–148,223 |
> | 4613 | 7676bF | 0.800 | 0.345 | 0.570 | 0.349 | 0.472 | |
> | 4649 | 7283bH* | 0.712 | 0.314 | 0.500 | 0.255 | 0.320 | 151,169–181,807 |
> | 4649 | 7283bF | 0.846 | 0.380 | 0.729 | 0.428 | 0.590 | |
>
> *Appendix 4—table 1 continued on next page*

*Appendix 4—table 1 continued*

| LOT | NEaar | Asx D/L | Glx D/L | Ala D/L | Val D/L | Ile A/I | Thermal age (years) |
|---|---|---|---|---|---|---|---|
| 4652 | 7286bH* | 0.678 | 0.277 | 0.463 | 0.233 | 0.302 | 168,899–202,088 |
| 4652 | 7286bF | 0.822 | 0.355 | 0.698 | 0.391 | 0.528 | |
| 4671 | 7316bH* | 0.681 | 0.270 | 0.468 | 0.242 | 0.299 | 166,094–198,795 |
| 4671 | 7316bF | 0.863 | 0.339 | 0.687 | 0.384 | 0.577 | |
| 4675 | 7320bH* | 0.687 | 0.279 | 0.494 | 0.279 | 0.330 | 192,147–230,607 |
| 4675 | 7320bF | 0.832 | 0.321 | 0.688 | 0.423 | 0.590 | |
| 4683 | 7328bH* | 0.764 | 0.392 | 0.676 | 0.373 | 0.505 | 378,398–467,602 |
| 4683 | 7328bF | 0.879 | 0.425 | 0.821 | 0.523 | 0.737 | |
| 4697 | 7342bH* | 0.752 | 0.388 | 0.653 | 0.368 | 0.497 | 378,398–467,602 |
| 4697 | 7342bF | 0.904 | 0.659 | 0.920 | 0.777 | 0.973 | |
| 14426 | 10581bH* | 0.730 | 0.870 | 0.855 | 0.855 | 0.985 | 3,238,624–4,188,207 |
| 14426 | 10581bF | 0.790 | 0.930 | 0.960 | 1.010 | 1.280 | |
| 15575 | 10955bH* | 0.860 | 1.035 | 0.981 | 1.001 | >1.2 | 14,387,543–18,460,416 |
| 15575 | 10955bF | 0.924 | 1.001 | 0.961 | 1.005 | >1.2 | |
| 15578 | 10958bH* | 0.891 | 1.040 | 0.984 | 1.007 | >1.2 | 14,387,543–18,460,416 |
| 15578 | 10958bF | 0.916 | 1.009 | 0.960 | 0.999 | >1.2 | |
| 15579 | 10959bH* | 0.811 | 1.021 | 0.965 | 1.007 | >1.2 | 14,387,543–18,460,416 |
| 15579 | 10959bF | 0.926 | 1.006 | 0.969 | 1.005 | >1.2 | |
| 15582 | 10962bH* | 0.881 | 1.033 | 0.976 | 1.012 | >1.2 | 14,387,543–18,460,416 |
| 15582 | 10962bF | 0.915 | 1.002 | 0.947 | 1.016 | >1.2 | |
| 13902 | 10573bH* | 0.965 | 1.050 | 0.925 | 1.12 | >1.2 | 13,764,246–18,893,425 |
| 13902 | 10573bF | 0.935 | 1.030 | 0.925 | 1.065 | >1.2 | |
| 13901 | 10574bH* | 0.920 | 1.040 | 0.945 | 1.16 | >1.2 | 8,943,148–11,841,107 |
| 13901 | 10574bF | 0.935 | 1.015 | 0.930 | 1.08 | >1.2 | |
| 13898 | 10575bH* | 0.945 | 1.050 | 0.930 | 1.17 | >1.2 | 14,746,875–20,367,942 |
| 13898 | 10575bF | 0.935 | 1.010 | 0.930 | 1.095 | >1.2 | |

**Appendix 4—table 2.** %FAA values (%FAA = [FAA]/[THAA] * 100). b = bleached. Total% FAA for Laetoli are calculated on the basis of Asx, Gly, Ala, Val only. * [Ala] and [Gly] > 100% are likely due to the effect of decomposition of other amino acids to FAA Gly and FAA Ala (e.g. Ser) (*Walton, 1998*). (*H*) Sample 4605 had been exposed to high temperatures in the burial environment (burning) (*Crisp 2013*).

| LOT | NEaar | Asx | Gly | Ala | Val | Ile | Average |
|---|---|---|---|---|---|---|---|
| 1868 | 6887 | 7 | 16 | 18 | 19 | 6 | 13 |
| 1872 | 6888 | 4 | 11 | 10 | 3 | 2 | 6 |
| 1866 | 6889 | 18 | 35 | 39 | 15 | 13 | 24 |
| 1849 | 6891 | 26 | 41 | 41 | 21 | 18 | 30 |
| 1850 | 6893 | 17 | 14 | 57 | 34 | 30 | 30 |
| 1823 | 6896 | 36 | 41 | 47 | 23 | 21 | 34 |
| 1819 | 6899 | 22 | 45 | 56 | 31 | 28 | 36 |
| 1840 | 6907 | 18 | 40 | 51 | 26 | 23 | 32 |
| 4605(*H*) | 7198 | 70 | 74 | 83 | 61 | 56 | 69 |
| 4613 | 7676 | 53 | 57 | 66 | 44 | 41 | 52 |

*Appendix 4—table 2 continued on next page*

*Appendix 4—table 2 continued*

| LOT | NEaar | Asx | Gly | Ala | Val | Ile | Average |
|---|---|---|---|---|---|---|---|
| 4649 | 7283 | 49 | 49 | 63 | 40 | 39 | 48 |
| 4652 | 7286 | 48 | 47 | 61 | 42 | 40 | 48 |
| 4671 | 7316 | 50 | 61 | 72 | 47 | 45 | 57 |
| 4675 | 7320 | 62 | 59 | 74 | 48 | 47 | 58 |
| 4683 | 7328 | 90 | 71 | 91 | 64 | 66 | 76 |
| 4697 | 7342 | 55 | 83 | 96 | 64 | 58 | 71 |
| 14426 | 10581 | 50 | 122* | 138* | 90 | 84 | 97 |
| 15575 | 10955 | 53 | 47 | 55 | 46 | n/a | 50 |
| 15578 | 10958 | 70 | 62 | 72 | 62 | n/a | 67 |
| 15579 | 10959 | 29 | 27 | 31 | 26 | n/a | 28 |
| 15582 | 10962 | 78 | 62 | 74 | 64 | n/a | 69 |
| 13902 | 10573 | 74 | 64 | 82 | 65 | n/a | 71 |
| 13901 | 10574 | 88 | 78 | 92 | 74 | n/a | 83 |
| 13898 | 10575 | 87 | 76 | 93 | 73 | n/a | 82 |

## B. Proteome persistence and patterns of degradation

### Fossil OES

Systematic analyses of the bleached proteomes surviving in sub-fossil OES shows that:

1. The complexity of the proteome decreases with increasing age of the sample (using the extent of racemization of Val, a slow racemizer, as a proxy for time).

2. The loss of proteins (number of identified product ion spectra; number of identified proteins) follows an exponential pattern over time.

3. The major proteins identified in modern OES are excellently preserved in samples up to 150,000 years old (Val THAA D/L ~0.5).

4. Beyond Val D/L ~0.5, quasi-exponential degradation of the proteome proceeds and only the main mineralization proteins, SCA-1 and SCA-2, are detected in the OES analysed from Wonderwerk and Laetoli.

5. The average length of the peptides identified appears to be stable (10–12 amino acids) across all samples, but the characteristics of the amino acid residues change drastically in the oldest samples, with mean hydropathicity index up to −17, due to the preferential preservation of 'DDDD'-containing peptides.

### Purified OES proteome heated at 140°C

The major proteins identified from the organic fraction extracted by demineralization of bleached OES and, once purified and lyophilized, heated at high temperature (140°C) for 2, 8, 24 and 120 hr are a subset of the sequences identified in modern and fossil OES powders. SCA-1 and SCA-2 are the dominant proteins, followed by the vitelline-membrane outer layer protein 1, von Willebrand factor, immunoglobulin and apolipoprotein D. SCA-1 and SCA-2 do not survive beyond 8 hr heating, while peptide sequences from other proteins were identified in samples heated for 24 hr. However, this is likely to be arising from contamination, due to lengthy and more complex sample preparation. No peptides were detected in the 120-hr heated sample.

*Figure 2—figure supplement 2(C–F)* shows that while degradation proceeds rapidly andfollows an exponential trajectory (as for fossil OES), the peptides surviving duringaccelerated diagenesis in water display increasing hydrophobicity, contrary to that

observed in OES fossil eggshell. This difference offers further support to the hypothesis that mineral binding of charged (acidic) peptides is key for survival.

**Appendix 4—table 3.** : Proteins identified in the purified extract from bleached OES, heated at 140°C in ultrapure water for 2, 8, 24 and 120 hr (identified by at least 2 unique peptide sequences). Values are number of peptide sequences identified.

| Protein description | 2 hr | 8 hr | 24 hr | 120 hr | Total |
|---|---|---|---|---|---|
| RecName: Full = Struthiocalcin-1; Short = SCA-1 | 295 | 21 | | | 316 |
| RecName: Full = Struthiocalcin-2; Short = SCA-2 | 124 | 8 | | | 132 |
| Vitelline membrane outer layer protein 1-like [*Struthio camelus australis*] | 22 | 2 | | | 24 |
| von Willebrand factor partial [*Struthio camelus australis*] | 20 | | | | 20 |
| immunonoglobulin heavy chain variable region partial [*Struthio camelus*] | 18 | | | | 18 |
| Apolipoprotein D [*Struthio camelus australis*] | 14 | 2 | | | 16 |
| Mitogen-activated protein kinase MLT [*Struthio camelus australis*] | | 7 | 9 | | 16 |
| Aggrecan core protein [*Struthio camelus australis*] | 13 | | | | 13 |
| iron binding protein [*Struthio camelus*] | 9 | | | | 9 |
| Histone H4 [*Struthio camelus australis*] | | 4 | 4 | | 8 |
| Cyclin-K [*Struthio camelus australis*] | 7 | | | | 7 |
| BPI fold-containing family B member 4 partial [*Struthio camelus australis*] | 4 | 2 | | | 6 |
| Histone H2B 1/2/3/4/6 [*Struthio camelus australis*] | | 3 | 3 | | 6 |
| Complement C3 [*Struthio camelus australis*] | | 2 | 3 | | 5 |
| PREDICTED: histone H2B 1/2/3/4/6 [*Struthio camelus australis*] | | 2 | 2 | | 4 |
| PREDICTED: polymeric immunoglobulin receptor [*Struthio camelus australis*] | 3 | | | | 3 |
| Carbonic anhydrase 4 partial [*Struthio camelus australis*] | 2 | | | | 2 |

## C. Persistence of SCA-1 and SCA-2

### Fossil OES

*Figure 3—figure supplement 1* displays the combined spectral count for the sequence of SCA-1. Unmodified peptides, as well as peptides with modifications, were considered for all samples. This allows us to visualize the frequency (intensity) at each amino acid position and to identify patterns of survival in time. It is obvious that the occurrence of diagenesis-induced modifications (e.g. deamidation) increases over time (see also *Supplementary file 3*), and that samples older than 150,000 years display a striking pattern of preservation, with the only low-frequency region detected consistently being the 'DDDD-' containing peptides. The main features of the protein structure also highlight that highly-structured regions are not preferentially preserved and that the DDDD-motif is part of a highly flexible (disordered) domain.

*Figure 3—figure supplement 2* shows the degradation pattern of the second major protein in OES: SCA-2. Similarly to SCA-1, modifications increase in frequency over time, and the overall frequency decreases. SCA-2 is however not detected with high confidence in samples older than 150,000 years old. Furthermore, the 150-ka old SCA shows two main regions surviving: one, around positions 45−55, contains an 'EEE' motif and broadly corresponds to peptide ASIHSEEEHQAIV, investigated computationally; the second, around positions 72–82, follows another E-rich region (corresponding to peptide SDSEEEAGEEVW investigated computationally) but contains instead a 'VWIG' motif that is highly conserved in all C lectins sequenced in biomineralized organisms (eggshell, sea-urchin spicules). By

analogy with the structure of SCA-1, it is likely that this region is helical. Therefore, this points towards two preservation mechanisms: mineral-binding ('EEE' region) and dehydration ('VWIG'). Dehydration is also observed (water loss) as PThigh frequency in this region, further supporting this hypothesis

## Extracted OES matrix heated at 140°C

SCA-1 and SCA-2 were not detected in OES matrix heated for 24 hr and 120 hr. Therefore here we show the results from the 2 hr and 8 hr heating experiments (*Figure 3—figure supplement 1* and *2*).

Around 50% of the sequence of SCA-1 survives 2 hr heating at 140°C in the absence of the mineral phase; the regions covered with higher frequency are: around residues 40−50, between two beta strands; and around residues 83–100, thus containing the D93-D96 motif. The dominant modification in this region, rather counter-intuitively, is water loss (−18.01 Da). By 8 hr heating this region is however almost disappeared; the AHLASIHT peptide (38–45) between the two alpha-helices is the most long-lived SCA-1 region when accelerated diagenesis takes place in water. SCA-2 appears to survive better during the experiment; most of the sequence can be detected after 8 hr heating, albeit with low spectral counts.

Overall, the patterns of degradation of SCA-1 and SCA-2 in water, as well as the signal detected from the whole OES matrix, are sufficiently different from those detected in fossil OES to support the hypothesis that different preservation mechanisms and degradation pathways operate in the OES and in aqueous environment (*Figure 3—figure supplement 1* and *2*). However, the survival of highly-acidic regions in SCA-1 combined with the frequency of observed dehydration might also support the idea that these peptides stick to the glass surface of the vial during the high-temperature experiments, and are thus protected from the water.

## Appendix 5

# Authenticity of the ancient sequences

## A. Independent replication

In order to authenticate the fossil peptide sequences obtained from the oldest OES examined in this study (from Laetoli) and to obtain any additional information on possible bias introduced by our sample preparation and analytical methods, a subsample of bleached powder from each OES (LOT 13901, 13902, 13898) was sent to the Proteomics Laboratory, Center for Geogenetics, University of Copenhagen. The matrix was re-extracted and analysed as detailed in the Materials and methods section.

The results from the three analyses were merged in a single MGF file and searched against the Struthioniformes database using the same parameters as described for all other OES samples considered in this study. For comparison, a search was also performed on all Laetoli subsamples analysed in Oxford, combined in PEAKS to yield a single output deriving from all fractions. The results are reported in and *Appendix 5—table 1*.

Two of us (JTO and BD) performed manual *de novo* identification of the sequences from raw product ion spectra without prior knowledge of the PEAKS assignments (*Supplementary file 2* reports all the raw spectra, manually annotated).

**Appendix 5—table 1.** Peptides identified in all Laetoli OES samples, prepared in York (analysed in Oxford) and Copenhagen. (*Gla) = this residue was found to be in the decarboxylated form, i.e. glutamate residues that have been post-translationally modified by vitamin K-dependent carboxylation to form gamma-carboxyglutamic acid (Gla), which binds calcium. We conducted manual *de novo* analyses of all product ion spectra and determined either complete or partial sequences for all of them, independently identifying sequences assigned (assisted *de novo*) by the software (PEAKS Studio).

| Sequence | Protein | Laboratory | Score | MS/MS count |
|---|---|---|---|---|
| AGAHLASIHTSEEHR | SCA-1 | Copenhagen | 47.14 | 4 |
| HYSALDDDDYPKGK | SCA-1 | Copenhagen | 35.69 | 2 |
| AGAHLASIH | SCA-1 | Copenhagen | 31.45 | 3 |
| ERNAFICK | SCA-1 | Copenhagen | 28.12 | 1 |
| GNCYGYFR | SCA-1 | Copenhagen | 28.1 | 1 |
| DVWIGLFR | SCA-1 | Copenhagen | 26.38 | 5 |
| ALDDDDYPK | SCA-1 | York | 39.39 | 28 |
| ALDDDDYPKG | SCA-1 | York | 41.34 | 14 |
| DDDDYPKGK | SCA-1 | York | 40.79 | 3 |
| DDDYPKGK | SCA-1 | York | 32.89 | 1 |
| HYSALDDDDYPK | SCA-1 | York | 51.09 | 1 |
| KHYSALDDDDYPK | SCA-1 | York | 34.86 | 2 |
| LDDDDYPK | SCA-1 | York | 34.35 | 12 |
| LDDDDYPKG | SCA-1 | York | 35.3 | 6 |
| LDDDDYPKGK | SCA-1 | York | 35.66 | 3 |
| SALDDDDYPK | SCA-1 | York | 41.04 | 10 |
| SALDDDDYPKG | SCA-1 | York | 39.15 | 5 |
| YSALDDDDYPK | SCA-1 | York | 34.46 | 3 |
| YSALDDDDYPKG | SCA-1 | York | 31.9 | 3 |
| RAEAWCR | SCA-1 | York | 30.65 | 1 |

*Appendix 5—table 1 continued on next page*

Appendix 5—table 1 continued

| Sequence | Protein | Laboratory | Score | MS/MS count |
|---|---|---|---|---|
| CYGFFPQELSWR | SCA-2 | Copenhagen | 30.98 | 1 |
| KPFICEYRT | SCA-2 | Copenhagen | 25.03 | 1 |
| GE(*Gla)EVWIGLHRPLGR | SCA-2 | York | 37.33 | 2 |
| LDYGSWYR | SCA-2 | York | 35.1 | 1 |
| AGE(*Gla)EVWIGLHRPLGR | SCA-2 | York | 34.64 | 2 |

The results in *Appendix 5—table 1* and *Supplementary file 2* show that:

a. 'DDDD'-containing peptides are found in samples prepared in Copenhagen, providing independent validation of the York results. However, the spectral count for these peptides is higher in the York set.

b. Only one extra SCA-1 peptide (RAEAWCR, 26–32) was found in the combined York samples, while five additional SCA-1 sequences (AGAHLASIHTSEEHR and AGAHLASIH, ERNAFICK, GNCYGYFR, DVWIGLFR) were identified in the Copenhagen set. We attribute this to different sample preparation protocols and to the larger sample size used in Copenhagen.

c. SCA-2 peptides CYGFFPQELSWR, KPFICEYRT were detected in the Copenhagen set, while GEEVWIGLHRPLGR/AGEEVWIGLHRPLGR and LDYGSWYR were detected in the York set. The spectral count was low in both sets.

## B. Amino acid analyses

All samples were analysed for chiral amino acids, and the concentrations were always well above the limit of detection (*Figure 4*). Low concentrations in the oldest samples would have indicated that contamination is likely even in a theoretically closed system, but our values for Wonderwerk, Olduvai and Laetoli are ~150–200 times higher than the typical blank values calculated on a normal RP-HPLC analytical run in the NEaar laboratory (University of York) (*Demarchi et al., 2011, 2015*; *Pierini et al., 2016*). The concentration values are typically affected by a larger error than D/L values due to calculation errors introduced during sample preparation, explaining the variability observed. The Laetoli samples retain between 50–70% of the amino acids measured in modern OES. While leaching from the carbonate matrix cannot be completely excluded, in laboratory diagenesis experiments this leaching amounts to only ~0.5% of the original THAA concentration in 72-hr bleached modern OES heated at high temperature (*Crisp et al., 2013*). Therefore, we attribute the observed loss to decomposition processes affecting the amino acids in these highly-degraded samples. Furthermore, all D/L values and% FAA (and *Appendix 4—table 1*and *2*) are consistent with the age of the samples, excluding the possibility that the proteins analysed are simply contamination.

## C. Carry-over analysis

The analysis of procedural blanks analysed before and after each sample is fundamental for evaluating the authenticity of the fossil sequences.

We detected low levels (low number of spectra) of actin, ubiquitin/polyubiquitin, aggrecan core protein, vitelline membrane proteins, iron-binding protein, serotransferrin, von Willebrand factor, BPI-fold containing family B member 4, tenascin, neuronal pentraxin, apolipoprotein D, ovomucoid, serum albumin, endoplasmin, neural proliferation differentiation and control protein 1 and tubulin in most of the LC-MS/MS blanks. More significantly, both SCA-1 and SCA-2 were detected (high sequence coverage, high number

of spectra) in blanks surrounding modern and fossil samples from Elands Bay Cave and Pinnacle Point (*Figure 4*). As SCA-1 and SCA-2 are the two major proteins in OES and the number of spectra in the blanks decreases with the age of the samples, this is indicative of carry-over, i.e. high-abundance peptides from a sample being retained in the LC column and being eluted during the successive blank run. This would obviously create a major issue for any claims of ancient sequences from fossil samples. Indeed, SCA sequences identified in blanks look remarkably similar to 'old' sequences (*Figure 4D*), because peptides with ion-binding characteristics (such as the 'DDDD'-containing peptide in SCA-1) would also exhibit equal affinity for the solid phase of the LC column. In order to exclude that the signal detected in ancient samples by LC-MS/MS may be due to carry-over we took the following precautions:

1. A procedural blank was run before and after each sample and the signal obtained for samples and blanks was compared directly in order to ensure that a 200x fold decrease in signal intensity was observed

2. Modern OES was analysed in November 2014; Elands Bay Cave and Pinnacle Point samples were analysed in December 2014

3. Subsamples from Laetoli were analysed in two separate batches, >4 months after the Elands Bay Cave/Pinnacle Point series: April 2015 - 13901E, 13902T; June 2015: 13901T and 13901N, 13902E, 13898E and 13898T. The results were replicated independently in the separate analyses.

4. Wonderwerk OES was analysed in August 2015.

5. Olduvai OES was analysed in March 2016.

A more refined analysis was carried out on the blanks in order to estimate the extent of carry-over, because spectral counts data only give limited quantitative information. Firstly, two representative younger fossil samples (Elands Bay Cave LOT 1823 and Elands Bay Cave LOT 1819) and the blank analysed between the two runs were considered. The total ion chromatogram (*Figure 4B*) shows an obvious reduction in signal intensity between the sample and the following blank.

The ion chromatogram (*Figure 4E*) for peptide LDDDDYPK was extracted and compared for the three samples. The signal between the first sample and the blank decreases by a factor of 1330, indicating that a similar further decrease is expected between the blank and the following sample. However, in the following sample (EBC_1819) the signal intensity increases by a factor of 1254 compared to the previous blank. Therefore, the extent of carry-over is negligible but detectable by modern mass spectrometers, indicating that a qualitative analysis is insufficient to address sample carry-over. The signal in the blank is just above the threshold at which a MS/MS of this peptide is triggered, therefore the likelihood of the further carried over peptide resulting in a MS/MS in the following (sample) run is extremely low. As a consequence, the LDDDDYPK peptides identified in sample EBC_1819 will all be genuine IDs.

For the Laetoli and Wonderwerk OES samples (Laetoli shown in *Figure 4F*), a more thorough investigation was carried out using the software Progenesis QI (nonlinear Dynamics, Waters) and generating XICs for all features in all samples after alignment in order to obtain an estimate of the relative abundance of each identified peptide across samples. The abundances of all 'DDDD'-containing peptides (derived from natural cleavage, and trypsin/elastase digestion) in the April-15 and June-15 batches were summed in order to compensate for the different ionization potential of different species. These were then compared across samples and blanks and the fold changes calculated. The calculated maximal carry-over was well below 1% and is likely to be lower due to poor signal to noise ratios especially in the blank runs. Assuming that the carry-over from a blank into the following sample is again 1%, the effective carry-over from sample to sample can

be estimated to be below 0.01%. Therefore, this is further supporting evidence for the authenticity of the peptide sequences in these samples.

## D. Damage patterns

Post translational modifications (PTMs) of fossil proteins can result from diagenesis processes and therefore the expected trend is one of increase with increasing age of the sample for comparable amounts of sequence preservation. Labile PTMs originally present on the molecule (e.g. phosphorylation) are however expected to be lost over time. Finally, preparation-induced modifications (e.g. carbamidomethylation) should be found in all samples, irrespective of their age, if that part of the sequence is intact. We examined the patterns of PTMs in SCA-1 and SCA-2 for all the samples analysed in this study. The graphs in *Supplementary file 3* show that the most common PTMs in bleached OES are:

- Carbamidomethylation (57.02 Da) on cysteines, as a consequence of sample preparation (reduction-alkylation of disulfide bridges)

- Deamidation (+0.98 Da) of Asn, Gln

- Decomposition of Arg to ornithine (−42.02 Da)

- Pyroglutamic acid formation from either N-terminal Gln or Glu by loss of ammonia/water (−17.01 Da, 18.01 Da)

- Oxidation of Met, Trp, His (single/double: 15.99 Da, 31.99 Da), or hydroxylation of Lys, Pro, Arg, Tyr (15.99 Da)

- Phosphorylation of Ser, Thr (79.97 Da)

- Amidation of the C-terminus (−0.98 Da)

Other modifications were also detected less frequently; the full list and position of each PTM is given in *Supplementary file 4*. Low spectral counts from sample 4613 (Pinnacle Point) result in lower PTM frequencies.

For SCA-1 diagenesis-induced modifications increase in frequency in fossil samples as compared with modern samples, and then decrease for progressively older Pleistocene samples due to loss of protein sequence preservation. This increases confidence in the authenticity of these older sequences reported. SCA-2 shows a very similar pattern: deamidation of Asn and Gln and oxidation of Met, Trp and His are clearly present throughout, and their frequency increases with age (*Supplementary file 3*).

## E. Volatiles

A number of peaks were present in the subfossil Laetoli sample that were not present in the blank, which corresponded with sulfur-containing VOCs in particular. The GC/QTOFMS has a mass resolution of <3 ppm, allowing extracted ion chromatograms to be used to isolate these sulfur-containing hydrocarbons from the complex VOC mix. The mass spectra across each suspected thiol / thioether peak were compared to the NIST MS library and the similarity and reverse fits for identified species are shown in *Appendix 5—table 2*. These values show how well the measured mass spectrum matches the library spectrum and are given out of 1000. Due to the low concentrations, even after background subtraction, some contaminant ions are still present, resulting in low similarity fits (comparison of the measured to library spectrum). However, by using the reverse fit (where the library spectrum is compared to the measured spectrum) much higher fits are achieved, since background contaminant ions are ignored. Therefore, the sulfur-containing VOCs have been assigned structures based on their accurate mass measurements, and whether the MS similarity or reverse fits are greater than 700. The Kovats retention index order of the

identified species is consistent with previous observations (NIST chemistry webbook), increasing the certainty in the identifications.

The volatile S-containing alkanes identified, absent in the procedural blank, are consistent with degradation of organic matter within a closed system under anoxic conditions, and would not be retained over these timescales within an open system. The calcium carbonate biomineral must therefore effectively form a closed system for the entrapped organic material, enabling predictable degradation. Analysis of the volatile fraction therefore confirmed both the extreme degradation observed by amino acid racemization and predicted based on the thermal age, but also highlighted the superiority of ostrich eggshell as a closed system.

**Appendix 5—table 2.** Crushed eggshell sulfur-containing VOC emissions from 2.7 Ma OES (Laetoli LOT 13901). NA = no structural isomer can be determined.

| Compound | Retention time (min) | Measured m/z | Chemical formula | NIST MS similarity | Reverse fit | |
|---|---|---|---|---|---|---|
| Ethane thiol/di-methylsulfide | 3.42 | 62.0196 | $C_2H_6S$ | NA | NA |  |
| Methylthioethane | 5.59 | 76.0354 | $C_3H_8S$ | 726 | 821 |  |
| Diethylsulfide | 8.24 | 90.0511 | $C_4H_{10}S$ | 725 | 834 |  |
| 1-methylthiopropane | 8.63 | 90.0509 | $C_4H_{10}S$ | 646 | 721 |  |
| Unknown isomer | 8.86 | 104.0663 | $C_5H_{12}S$ | NA | NA | NA |
| 2-ethylthio-propane | 9.66 | 104.0664 | $C_5H_{12}S$ | 535 | 723 |  |
| 2-methyl-1-(methylthio)-propane | 10.26 | 104.0670 | $C_5H_{12}S$ | 636 | 798 |  |
| ethylpropylsulfide | 10.85 | 104.0671 | $C_5H_{12}S$ | 628 | 801 |  |
| 2-methyl-2-(methylthio)-butane | 11.89 | 118.0823 | $C_6H_{14}S$ | 567 | 777 |  |
| 1-(ethylthio)-2-methyl-propane | 12.18 | 118.0824 | $C_6H_{14}S$ | 817 | 856 |  |
| 2-methyl-3-(methylthio)-butane | 12.286 | 118.0823 | $C_6H_{14}S$ | 600 | 820 |  |
| Unknown isomer | 12.791 | 118.0826 | $C_6H_{14}S$ | NA | NA | |
| 3-methyl-1-(methylthio)-butane | 12.821 | 118.0827 | $C_6H_{14}S$ | 779 | 872 |  |

*Appendix 5—table 2 continued on next page*

*Appendix 5—table 2 continued*

| Compound | Retention time (min) | Measured m/z | Chemical formula | NIST MS similarity | Reverse fit | |
|---|---|---|---|---|---|---|
| Tetrahydro-2,5-di-methylthiophene | 13.06 | 116.0670 | $C_6H_{12}S$ | 657 | 741 | 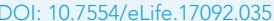 |
| Tetrahydro-2,5-di-methylthiophene | 13.14 | 116.0670 | $C_6H_{12}S$ | 622 | 707 | |

## F. Ancient DNA

A single-stranded library was built (***Gansauge and Meyer, 2013***) and sequenced on an Illumina HiSeq 2500. Of the 282,924 reads sequenced, 68,204 remained after quality filtering and filtering for reads larger than 30 bp using AdapterRemoval (***Lindgreen, 2012***). Remaining reads were mapped to an ostrich reference nuclear and mitochondrial genomes (***Zhang et al., 2015***) using BWA-aln (Version: 0.7.5a-r405) with default settings (***Li and Durbin, 2009***) and the results examined using SAMtools (***Li et al., 2009***). No sequences were successfully aligned to the reference genomes.

