## [Decision Letter]

Thank you for submitting your article "Protein sequences bound to mineral surfaces persist into deep time" for consideration by *eLife*. Your article has been reviewed by three peer reviewers, and the evaluation has been overseen by a Reviewing Editor and Diethard Tautz as the Senior Editor. The following individual involved in review of your submission has agreed to reveal his identity: Michael Richards (Reviewer #1).

The reviewers have discussed the reviews with one another and the Reviewing Editor has drafted this decision to help you prepare a revised submission.

Summary:

This is a ground-breaking paper that shows, through a range of methods, survival of a protein in eggshell that is approximately 3.8 million years old. Molecular dynamics simulations are employed as a virtual probe of calcite-protein interactions.

Essential revisions:

While the data and analysis are quite strong, the manuscript is hard to follow and should be extensively rewritten taking care to introduce each disparate area using words that are understandable across disciplines. The lack of words in the article and the extremely heavy data combined with the disparate research areas make it challenging to read. Explanation of the interface between the various sets of data should also be made more clear. The authors should also think about the 'evidence' for some of their claims and their reference to particular sets of data throughout the manuscript and the various supplementary files as this is not consistent.

The primary result is demonstrated convincingly and states that the protein domain with the greatest binding affinity is the best preserved in the fossilized material. The mechanism of protein stabilization is "explained by entropy loss at the mineral surface" which the authors argue lowers "the effective temperature of the local environment". In the reviewer's opinion the data support the main conclusion of the paper; however, the mechanistic interpretation is somewhat problematic for the following reasons:

1) The driving force for the stabilization of the protein-mineral interaction is presumed to be entropy loss. Certainly one would expect both the translational and rotational degrees of freedom (of both water and the biomolecule) to be inhibited upon interaction with the surface that would be reflected in a smaller TS term in the free energy of binding. However, there is also be an enthalpic contribution to the free energy. While the authors calculate the enthalpic component at constant volume (i.e. using internal energies) they only estimate the TS term and cannot be certain that TS dominates over H. The choice of TS over H is not well justified as TS was not determined.

2) While entropy loss may be the driving force for protein adsorption from the aqueous environment, once the egg shell is formed these proteins find themselves in contact with mineral rather than a bulk fluid phase. Therefore, the entropy loss that drove the protein to bind to the mineral doesn't appear to be at all relevant to the longevity of the protein in the shell where bulk water is not generally present.

3) The somewhat inhibited rates of diffusion and water exchange adjacent to the adsorbed biomolecules are cited as an indication that entropy loss is an important factor. If the average surface bound water has a residence time of ~120 ps in the absence of protein (with no uncertainties provided), the increase to 123-135 psec in the presence of protein doesn't seem overwhelmingly significant. Also, per the discussion above (1), the somewhat inhibited water dynamics could also be caused in part by enthalpic interactions between water and the surface bound biomolecule.

---

## [Author Response]

*Essential revisions:*

*While the data and analysis are quite strong, the manuscript is hard to follow and should be extensively rewritten taking care to introduce each disparate area using words that are understandable across disciplines. The lack of words in the article and the extremely heavy data combined with the disparate research areas make it challenging to read. Explanation of the interface between the various sets of data should also be made more clear. The authors should also think about the 'evidence' for some of their claims and their reference to particular sets of data throughout the manuscript and the various supplementary files as this is not consistent.*

We agree that the paper attempted to be too concise at the expense of readability, especially when combining evidence from and the language of different disciplines. The text has been integrated and explanations of the interface between datasets have been improved, drawing from information that was previously contained in the supplementary files. The additional text is marked as underlined in the “tracked changes manuscript file”.

*The primary result is demonstrated convincingly and states that the protein domain with the greatest binding affinity is the best preserved in the fossilized material. The mechanism of protein stabilization is "explained by entropy loss at the mineral surface" which the authors argue lowers "the effective temperature of the local environment". In the reviewer's opinion the data support the main conclusion of the paper; however, the mechanistic interpretation is somewhat problematic for the following reasons:*

*1) The driving force for the stabilization of the protein-mineral interaction is presumed to be entropy loss. Certainly one would expect both the translational and rotational degrees of freedom (of both water and the biomolecule) to be inhibited upon interaction with the surface that would be reflected in a smaller TS term in the free energy of binding. However, there is also be an enthalpic contribution to the free energy. While the authors calculate the enthalpic component at constant volume (i.e. using internal energies) they only estimate the TS term and cannot be certain that TS dominates over H. The choice of TS over H is not well justified as TS was not determined.*

*2) While entropy loss may be the driving force for protein adsorption from the aqueous environment, once the egg shell is formed these proteins find themselves in contact with mineral rather than a bulk fluid phase. Therefore, the entropy loss that drove the protein to bind to the mineral doesn't appear to be at all relevant to the longevity of the protein in the shell where bulk water is not generally present.*

We thank the reviewers for these comments. We believe that the points raised stem from the fact that we did not make sufficiently clear in the original text that the driving force for protein adsorption is enthalpy (estimated using the internal energy) rather than entropy. The binding energies (calculated from the configurational energy) are all significantly negative, i.e. binding is more energetically favourable for the peptides than staying in solution. We argue that this increases the energy barrier required to achieve peptide degradation and therefore explains our experimental findings (i.e. the exceptional preservation of peptide sequences). In order to increase clarity a more extended explanation is now included in the text, considering the effect of binding on both the entropic and enthalpic components of the binding energy.

We discuss the inclusion of an entropy correction based on the change in water entropy alone because this shows an entropy gain for binding, due to the displacement of water molecules tightly bound to the calcite surface. These water molecules will gain entropy as they leave the tightly bound restricted surface and enter the bulk water where they can move much more freely. There will be an entropy loss for the peptide molecule as it becomes bound to the surface since this will restrict its translational freedom. This is unlikely to be as large as the entropy gain of ~20 water molecules displaced from the surface. The entropy term will be similar for both molecules as the binding configurations are similar, therefore we are confident that binding is favourable. We have modified the text to make this clearer to the reader and we have now provided the uncertainties for the values quoted in Table 2.

*3) The somewhat inhibited rates of diffusion and water exchange adjacent to the adsorbed biomolecules are cited as an indication that entropy loss is an important factor. If the average surface bound water has a residence time of ~120 ps in the absence of protein (with no uncertainties provided), the increase to 123-135 psec in the presence of protein doesn't seem overwhelmingly significant. Also, per the discussion above (1), the somewhat inhibited water dynamics could also be caused in part by enthalpic interactions between water and the surface bound biomolecule.*

We had previously focused our discussion on the entropic component and on temperature in order to link this to thermal age and degradation. We have now modified this discussion to concentrate on the concept of the energy barrier towards peptide hydrolysis and degradation, since this will hopefully be clearer for the reader. Therefore we have removed the specific reference to a given temperature. We were attempting to convey that water at the surface (that must be part of the peptide hydrolysis since the peptide is at the surface) is tightly bound – which means it will require more energy to overcome the barrier for hydrolysis. Since more energy is required this implies a greater ambient temperature will be required to degrade these peptides or, in other words, that the molecules behave as though they were “cooler” at the surface.